# HUMAN3R: EVERYONE EVERYWHERE ALL AT ONCE

**Yue Chen**[1,2] **Xingyu Chen**[1,2] **Yuxuan Xue**[3] **Anpei Chen**[2] **Yuliang Xiu**[2†] **Gerard Pons-Moll**[3,4]

[1]Zhejiang University  [2]Westlake University  [3]University of Tübingen, Tübingen AI Center
[4]Max Planck Institute for Informatics  [†]Corresponding Author

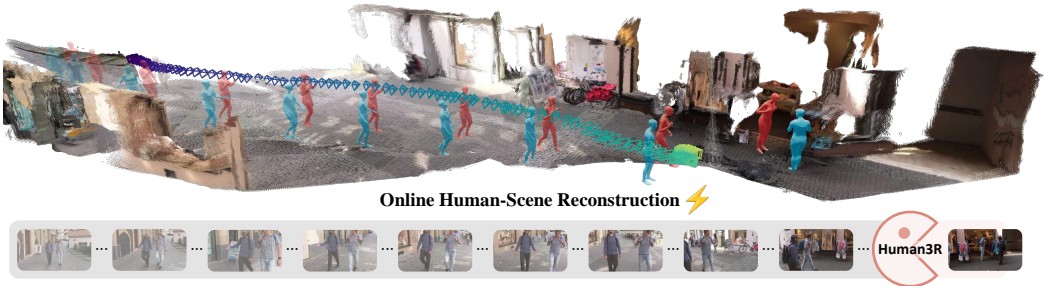

Figure 1: Given a stream of RGB images as input, Human3R enables human-scene reconstruction in an online, continuous manner, estimating global multi-person meshes, camera parameters, and dense scene geometry with each incoming frame in real time.

## ABSTRACT

We present Human3R, a unified, feed-forward framework for online 4D human-scene reconstruction, in the world frame, from casually captured monocular videos. Unlike previous approaches that rely on multi-stage pipelines, iterative contact-aware refinement between humans and scenes, and heavy dependencies, e.g., human detection, depth estimation, and SLAM pre-processing, Human3R jointly recovers global multi-person SMPL-X bodies (*"everyone"*), dense 3D scene (*"everywhere"*), and camera trajectories in a single forward pass (*"all-at-once"*). Our method builds upon the 4D online reconstruction model CUT3R, and uses parameter-efficient visual prompt tuning, to strive to preserve CUT3R's rich spatiotemporal priors, while enabling direct readout of multiple SMPL-X bodies. Human3R is a **unified model** that eliminates heavy dependencies and iterative refinement. After being trained on the relatively small-scale synthetic dataset BEDLAM for just **one day** on **one GPU**, it achieves superior performance with remarkable efficiency: it reconstructs multiple humans in a **one-shot** manner, along with 3D scenes, in **one stage**, in real-time (15 FPS) with a low memory footprint (8 GB). Extensive experiments demonstrate that Human3R delivers state-of-the-art or competitive performance across tasks, including global human motion estimation, local human mesh recovery, video depth estimation, and camera pose estimation, with a single unified model. We hope that Human3R will serve as a simple yet strong baseline, which can be easily adapted for downstream applications. Code, models and 4D interactive demos are available at `fanegg.github.io/Human3R`.

## 1 INTRODUCTION

Humans do not exist in isolation but constantly move in, interact with, and manipulate the world around us. Thus, understanding human behaviors requires putting them within a 3D world context, ideally in an online manner, as indicated in Fig. 2. In the field of 3D vision, this necessitates the 3D reconstruction of both **global human motions** and the **surrounding scene** from visual data [30], which is challenging, but fundamental for various downstream applications, including AR/VR, autonomous navigation, humanoid policy learning, and human-scene interaction.

Prior global human motion estimators typically follow one of two strategies: 1) *directly* estimating the global human motions aided with learned motion priors [72, 110]; 2) transforming human motion to world coordinates with SLAM-based [91] estimated global camera [44, 48, 80, 82, 89, 99, 109]. Considering the surrounding 3D scene, which is crucial for contextualizing human actions, recent advances attempt to jointly reconstruct 3D humans, scene, and cameras, either from multi-view images [16, 56, 74] or monocular videos [53].

However, these methods have two main limitations:
**1) Multi-stage/model/shot:** They [53, 56] reconstruct the scene and humans separately, then jointly refine them under contact constraints. The entire pipeline takes hours. In addition, a top-down multi-person mesh regressor is used, which requires off-the-shelf human detection and human tracking models [15, 41, 50, 71, 73, 106] to crop and associate each person before feeding into the single-person mesh regressor [31, 99], thus the inference speed considerably drops for images with multiple people.

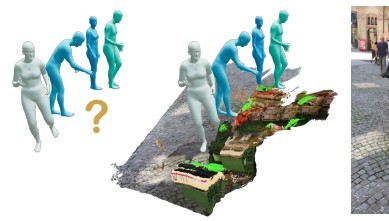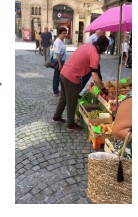

(a) w/o vs. w/ scene context      (b) Raw capture

Figure 2: Human behaviors (i.e., grocery shopping) become clearer when viewed within their surrounding environment.

**2) Heavy dependencies:** Apart from the modules mentioned above, numerous off-the-shelf dependencies are needed to preprocess the input images, including but not limited to metric depth estimators [4], generic 3D reconstruction models [24, 46, 91, 98] to obtain the 3D scene pointcloud, camera pose and intrinsics. Both limitations hinder real-time online inference, end-to-end learning, effortless deployment, and scalability to long sequences. We seek a unified one-stop solution.

We introduce Human3R, an *all-at-once* model for 4D human-scene reconstruction. The term "*all-at-once*" reflects several key aspects: 1) **One Model**: A unified model jointly reasons about humans, scene, and camera, rather than relying on separate off-the-shelf models for each component. 2) **One Stage**: In contrast to prior work with iterative refinement, our method runs in an online fashion. Specifically, our lightweight model operates on streaming video at real-time speeds (15 FPS on an RTX 4090) while maintaining competitive performance. It also offers configurable scalability, allowing for higher-fidelity reconstruction with larger backbones. 3) **One Shot**: With a bottom-up multi-person SMPL-X regressor, our model can reconstruct multiple persons in a single forward pass. 4) **One GPU, One Day**: Our model is parameter efficient, requiring only one day of training on a single NVIDIA 48GB GPU, still yielding state-of-the-art performance.

The main challenge in building such a unified model lies in the lack of large-scale video datasets with reliable annotations of global human motion, 3D scene, and camera pose. Existing real datasets [21, 33, 39, 94] are limited in scale, while synthetic ones, like BEDLAM [5], are limited in scene variations. Our key idea is to leverage the strong spatiotemporal priors [13, 27, 108] learned by a 4D reconstruction foundation model [97], and extend it through minimal tuning on a relatively small-scale human-scene dataset, achieving both data and parameter efficiency. This approach enables us to advance from point-only reconstruction to the joint reconstruction of dense scene point clouds and sequential SMPL-X body meshes [64] for multiple individuals in the scene.

Specifically, we build upon CUT3R [97], a recurrent 4D reconstruction foundation model for online metric-scale reconstruction, which maintains a persistent internal state that encodes *everywhere and everyone*, and incrementally updates it with new observations. We finetune CUT3R via visual prompt tuning (VPT) [35], with minimal learnable parameters prepended into the input space while the entire CUT3R backbone is kept frozen. BEDLAM [5] serves as our training data, which is small-scale yet high-quality, with 6k sequences featuring 3D scene depth, camera poses, and SMPL-X meshes of multiple persons in the world coordinates. Instead of naively prepending random initialized learnable tokens as visual prompts [35], we detect the human head tokens from CUT3R's image feature, complement it with human prior tokens [3] learned from human-specific datasets, and project them to *human prompts* using a learnable MLP, as shown in Fig. 4.

Our proposed *human prompts* are highly informative, as the head is the most discriminative keypoint on human bodies [3, 116]. As anchors (i.e., SMPL-X queries), these human prompts provide strong spatial priors for localizing and reconstructing the full human body. They self-attend to image tokens for spatial whole-body information aggregation, and cross-attend to the persistent internal state to make 3D human estimates scene-aware. Remarkably, Fig. 9 shows that the 3D scene reconstruction is also improved after finetuning for human reconstruction, demonstrating the mutual benefits of joint reasoning about humans and scene.

Simple yet effective, Human3R leverages the spatial and temporal priors learned by CUT3R to reason about humans, scene, and camera in a unified framework, efficiently processes long sequences with linear computational complexity (8 GB GPU memory footprint, 15 FPS inference speed), and supports scalable sequence length (thousands of frames) beyond training length (4 frames) by simply rolling out the state. Across various 4D tasks — including video depth estimation, camera pose estimation, human mesh recovery, and global human motion estimation — our method achieves superior performance over task-specific baselines while offering a unified and real-time solution.

## 2 RELATED WORKS

**Local Human Mesh Recovery.** Previous works on human mesh recovery (HMR) primarily focus on estimating the pose and shape parameters of a parametric body model, like SMPL [54], SMPL-X [64], and GHUM [105], *in the camera frame*. Early optimization-based methods fit SMPL model to IMU trajectories [94, 103] or to 2D landmarks by minimizing reprojection errors [6, 65]. In contrast, learning-based approaches, trained on large-scale image-body pairs, can regress SMPL parameters from images [37, 59] in a single pass. Progress in this field spans improvements in network architectures [25, 31, 52, 111], training and testing paradigms [19, 45, 76], kinematics designs [47, 49], camera models [43, 62], datasets [5, 25, 29, 36, 63], expressive body models [18, 28, 49, 64, 112], temporal consistency [17, 38, 42], and etc. For multi-person scenarios, most prior works adopt a top-down multi-stage approach: detect and crop each person before running single-person HMR. This is computationally expensive, scales poorly with more people, and often fails in crowded scenes due to occlusion and truncation. To overcome this, bottom-up methods [3, 87, 88, 100] recover multiple human meshes from a full image in one-shot scheme. Multi-HMR, for example, finetunes DINOv2 [60] on synthetic datasets [5, 63], and achieves strong performance. Our goal is even more ambitious: to reconstruct both the 3D scene and multiple humans *in the world frame* from monocular videos, using one unified model, in one forward pass, and in real-time.

**Global Human Motion Estimation.** Reconstructing world-grounded humans from long video sequences is an ill-posed problem, typically requiring additional priors or constraints. GLAMR [110] leverages the learned motion prior HuMoR [72] to infill occluded human motions and directly predict global trajectories from them. With SLAM (Simultaneous Localization and Mapping) [91], world-frame camera poses can be estimated, allowing local human meshes – recovered via HMR – to be transformed into the world frame [48, 109]. TRAM [99] robustifies and metrifies SLAM's camera estimation via masking the dynamic regions and estimating metric depth via ZoeDepth [4], which then serve as a reference frame to recover the

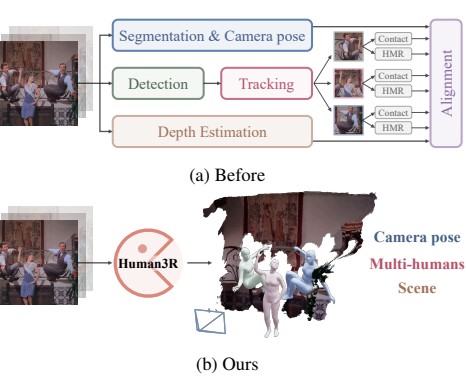

(a) Before

(b) Ours

Figure 3: Multi-stage vs. One-stage.

global human motion. GVHMR [80] introduces gravity and view-in direction constraints to further stabilize global human motions. Beyond these offline solutions, several online methods [82, 89] recurrently reconstruct global human meshes, maintaining consistently low memory and computation costs as the number of input frames increases. However, even excluding the SLAM step, most of these approaches still depend on multiple off-the-shelf estimators – such as human detection [48, 80, 82, 99, 109, 110], tracking [48, 80, 82, 99, 109, 110], segmentation [99], 2D keypoint detection [48, 80, 82, 109, 110], optical flow [89], camera-frame HMR [48, 99, 109, 110], and etc. Synchronization barriers between these branches often lead to cumulative errors and high computational overhead. In contrast, Human3R is an *all-in-one* model that not only online recovers human motions and root trajectories in the world frame, but also simultaneously reconstructs the surrounding 3D scene and estimates camera motions – an versatile framework not explored in prior works.

**Human-Scene Reconstruction.** Existing methods for joint human-scene 3D reconstruction typically perform global optimization over camera poses, pre-reconstructed scenes [24, 51, 79, 98] (please checkout related work about generic 3D reconstruction in Appendix B of *Sup.Mat.*), and SMPL mesh parameters inferred from multi-view images [56, 66], often regularized by learned motion priors [2, 53, 115]. Recently, optimization-free approaches have emerged: HAMSt3R [74], for example, jointly reconstructs the scene and DensePose [32] from multi-view images in a feed-forward manner, then fits SMPL meshes to the DensePose outputs. The most relevant work, JOSH3R [53], jointly reconstructs scene and human meshes from monocular videos with dynamic humans, but depends on camera-frame human meshes, detection, segmentation, and tracking, limiting scalability and efficiency. We eliminate all these dependencies, resulting in a lightweight yet unified model that directly predicts metric-scale dense scenes, global human motions, and camera poses from monocular video in a single forward pass. This unified approach distinguishes our method from previous works and opens up new possibilities for real-time applications in humanoid policy learning, autonomous navigation, and human-robot interaction.

## 3 METHODS

Our approach operates on a continuous stream of images in an online manner. At each timestep $t$, given an input image $\mathbf{I}_t \in \mathbb{R}^{W \times H \times 3}$, our goal is to estimate: 1) a set of $N$ human meshes $\{\mathbf{M}_t^n \in \mathbb{R}^{V \times 3}\}_{n=1}^N$ in the world coordinate system, where each $\mathbf{M}_t^n$ is parameterized by the SMPL-X body model with $V = 10{,}475$ vertices and $K = 54$ joints; 2) the camera extrinsic pose $\mathbf{T}_t \in \mathbb{R}^{3 \times 4}$, and intrinsic $\mathbf{C}_t \in \mathbb{R}^{3 \times 3}$; 3) the canonical point cloud $\mathbf{X}_t \in \mathbb{R}^{W \times H \times 3}$. Our feedforward inference operates online in real time. We first introduce preliminaries of the 3D human parametric model and the 4D reconstruction foundation model CUT3R [97] in Section 3.1. Then, in Section 3.2, we describe our proposed Human3R, which fine-tunes CUT3R to regress SMPL-X parameters for multiple 3D human bodies.

### 3.1 PRELIMINARIES

**Human Mesh Representation – SMPL-X [64].** We represent the 3D human body with the SMPL-X [54, 64], which is a low-dimensional parametric model of the human body mesh. Given the parameters of the local human pose (relative axis-angle rotations) $\boldsymbol{\theta} \in \mathbb{R}^{52 \times 3}$, body shape $\boldsymbol{\beta} \in \mathbb{R}^{10}$, facial expression $\boldsymbol{\alpha} \in \mathbb{R}^{10}$, and global human root transformation $\mathbf{P} = [\mathbf{R} \mid \mathbf{t}] \in \mathrm{SE}(3)$ parametrized by global orientation $\mathbf{R} \in \mathrm{SO}(3)$ and global translation $\mathbf{t} \in \mathbb{R}^3$, it outputs an expressive 3D human mesh $\mathbf{M}_t^n \in \mathbb{R}^{V \times 3}$, with $V = 10{,}475$ vertices. For brevity, we omit the timestep subscript $t$ and the id superscript $n$, as $\mathbf{M}_t^n \rightarrow \mathbf{M}$:

$$\begin{aligned} \mathbf{M} &= \text{SMPL-X}(\boldsymbol{\theta}, \boldsymbol{\beta}, \boldsymbol{\alpha}, \mathbf{P}) \\ \mathbf{P} &= \mathbf{T}\mathbf{P}^{\text{cam}} \end{aligned} \tag{1}$$

where the global human root transformation $\mathbf{P}$, in the world frame, is decomposed into the camera pose $\mathbf{T}$ and the local root transformation $\mathbf{P}^{\text{cam}}$ in the camera frame.

**4D Reconstruction Foundation Model – CUT3R [97].** To overcome the scarcity of world-grounded 4D human-scene datasets, we exploit the 4D reconstruction foundation model CUT3R [97], which is 4D-aware, and encodes rich 4D priors of real-world dynamics, including both scene (*everywhere*) and human (*everyone*), learned from large-scale 3D point cloud datasets. However, instead of explicitly separating the unstructured point clouds of humans from the scene, Human3R directly reads out global human bodies.

CUT3R performs recurrent reconstruction of metric-scale point maps (pixel-aligned point clouds in the world coordinate system) and camera poses in an online fashion, maintaining a fixed-size memory state that encodes *everything* that camera captures. This state enables the retrieval of past observations, while being continuously updated with new observations. Specifically, to transform a current image $\mathbf{I}_t$ into pixel-aligned point maps, the input image is encoded into a set of image tokens $\mathbf{F}_t \in \mathbb{R}^{(h \times w) \times c}$ through the ViT image tokenizer [23]: $\mathbf{F}_t = \text{Encoder}(\mathbf{I}_t)$. The image tokens then interact with the state in the following formulation:

$$[\mathbf{F}_t', \mathbf{z}_t'], \mathbf{S}_t = \text{Decoders}([\mathbf{F}_t, \mathbf{z}], \mathbf{S}_{t-1}) \tag{2}$$

where the init state representation is represented as a set of tokens $\mathbf{S}_0 \in \mathbb{R}^{768 \times 768}$, which are learnable parameters and are shared by all scenes. As the set of image tokens $\mathbf{F}_t$ is fed into the decoder, the previous state $\mathbf{S}_{t-1}$ is updated with new observations to produce an updated state $\mathbf{S}_t$, which encodes the spatial and temporal history of the scene, namely "context". Then, through the decoder, the image token $\mathbf{F}_t$ and camera token $\mathbf{z}_t$, attend with the context in current state $\mathbf{S}_t$, will be refined as $\mathbf{F}_t'$ and $\mathbf{z}_t'$. The camera token, designed to capture the image-level ego motion related to the scene, is prepended to the image tokens and is initialized as a learnable parameter $\mathbf{z}$. This bidirectional state-token interaction is implemented using two interconnected transformer decoders [98, 101, 102].

After the state-token interaction, the corresponding pixel-aligned metric scale (i.e., meters) 3D pointmaps in the camera and world coordinate systems are extracted via dense prediction head [70]: $\mathbf{X}_t^{\text{cam}} = \text{Head}_{\text{cam}}(\mathbf{F}_t')$, $\mathbf{X}_t^{\text{world}} = \text{Head}_{\text{world}}(\mathbf{F}_t', \mathbf{z}_t')$. The camera pose $\mathbf{T}_t$ is then regressed from camera tokens through an MLP network: $\mathbf{T}_t = \text{Head}_{\text{pose}}(\mathbf{z}_t')$, and the camera intrinsic $\mathbf{C}_t$ is solved using Weiszfeld [68] algorithms with predicted pointmaps, respectively.

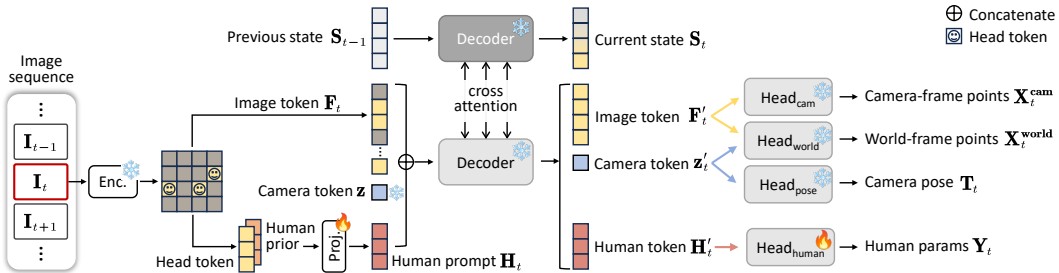

Figure 4: **Method Overview.** Human3R enables online human-scene reconstruction from video streams. Each frame is encoded into image tokens, with patch-level detection. Each detected head token is concatenated with a human prior token—sampled from a separate Multi-HMR [3] ViT-DINO encoder at the corresponding pixel coordinates—and subsequently projected into a *human prompt* $\mathbf{H}_t$. The *human prompts* serve as discriminative human-ID queries for the decoder: they self-attend with image tokens to aggregate spatial whole-body information and cross-attend with the scene state to retrieve temporally consistent human tokens within the 3D scene context. Only human-related layers are fine-tuned, other parameters remain frozen and are initialized from CUT3R [97].

## 3.2 HUMAN3R

**One-stage Global Human-Scene Reconstruction.** To preserve the rich 4D priors encoded by CUT3R, we adopt parameter-efficient visual prompt tuning (VPT) [35] for fine-tuning. Specifically, we introduce a small set of trainable parameters – prepended as visual prompts into the input space – to enable the readout of global human meshes, while keeping the entire CUT3R backbone frozen.

Unlike standard VPT, where additional parameters are randomly initialized learnable tokens, we instead detect human head tokens and transform them into *human prompts* using learnable projection layers. Specifically, we follow previous work [3] to detect the human head (defined by the head joint of SMPL-X model) as the primary keypoint of human. For each

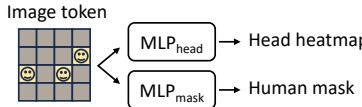

Figure 5: Detection and Segmentation.

patch index $(i, j) \in \{1, \dots, h\} \times \{1, \dots, w\}$, we predict whether the patch $\mathbf{u}^{i,j}$ contains the primary keypoint by computing a confidence score from the associated image feature token $\mathbf{F}^{i,j} \in \mathbb{R}^c$ using an MLP followed by a sigmoid activation $\sigma(\cdot)$, formulated as $s^{i,j} = \sigma\left(\text{MLP}_{\text{head}}(\mathbf{F}^{i,j})\right)$. We apply a threshold $\tau$ on $s^{i,j}$ to collect detected head token indexes, denoted as $\left\{\mathbf{u}^{i,j} \mid s^{i,j} \geq \tau\right\}_n$. We then predict the human mesh parameters $\mathbf{Y}_t = \{(\boldsymbol{\theta}, \boldsymbol{\beta}, \boldsymbol{\alpha}, \mathbf{P}^{\text{cam}})_t\}_n$ for all people with detected head tokens $\mathbf{F}_t^{\mathbf{u}} = \{\mathbf{F}_t^{i,j} \mid (i, j) \in \{\mathbf{u}_t\}_n\}$ in parallel:

$$\mathbf{H}_t = \text{Head}_{\text{projection}}(\mathbf{F}_t^{\mathbf{u}})$$
$$[\mathbf{F}_t', \mathbf{z}_t', \mathbf{H}_t'], \mathbf{S}_t = \text{Decoders}([\mathbf{F}_t, \mathbf{z}, \mathbf{H}_t], \mathbf{S}_{t-1}) \quad (3)$$
$$\mathbf{Y}_t = \text{Head}_{\text{human}}(\mathbf{H}_t')$$

where *human prompts* $\mathbf{H}_t$ is transformed from detected head tokens $\mathbf{F}_t^{\mathbf{u}}$ via the projection MLP, and the *SMPL-X parameters* $\mathbf{Y}_t$ are predicted by the human MLP from the refined human token $\mathbf{H}_t'$. $\mathbf{H}_t$ is inserted into the input space of the decoder. The colors ● and ● indicate learnable and frozen parameters, respectively. During fine-tuning, only the human-related MLP layers are updated, while all other parameters remain frozen. The *human prompts* serve as discriminative human ID queries: they self-attend with image tokens to aggregate spatial whole-body information and cross-attend with the scene state to retrieve temporal SMPL-X mesh parameters within the 3D scene context.

**Human Prior.** In practice, we found that CUT3R, trained on large-scale scene-centric datasets, lacks detailed human priors, leading to suboptimal performance in reconstructing fine-grained human poses and shapes. Thus, we enhance the head tokens $\mathbf{F}^{\mathbf{u}}$ with extra human-specific features from a human-centric image encoder. Particularly, we use another image tokenizer, the Multi-HMR [3] ViT image encoder, denoted as $\text{Encoder}_{\text{HMR}}$, which fully fine-tuned the pretrained DINO [10, 60] on human-specific

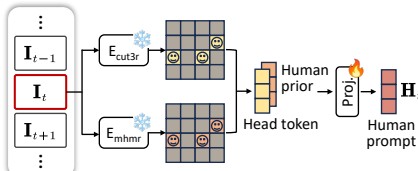

Figure 6: Integration of human priors via the human-centric Multi-HMR ViT-DINO image encoder.

datasets. Same as previous index-based query, we still use $\{\mathbf{u}\}_n$ to obtain the corresponding Multi-HMR ViT image tokens $\mathbf{F}_{\text{HMR}} = \text{Encoder}_{\text{HMR}}(\mathbf{I})$, to produce $\mathbf{F}_{\text{HMR}}^{\mathbf{u}} = \{\mathbf{F}_{\text{HMR}}^{i,j} \mid (i, j) \in \{\mathbf{u}\}_n\}$, which are subsequently concatenated with CUT3R head tokens $\mathbf{F}^{\mathbf{u}}$ and translated into *human prompts* by the projection MLP as: $\mathbf{H} = \text{Head}_{\text{projection}}(\mathbf{F}^{\mathbf{u}} \oplus \mathbf{F}_{\text{HMR}}^{\mathbf{u}})$, where $\oplus$ denotes concatenation along the channel axis. Notably, $\text{Encoder}_{\text{HMR}}$ is frozen during training. Concatenating Multi-HMR and CUT3R head tokens injects detailed human priors for improved body pose and shape prediction. And with additional *training-free* designs, Human3R also supports human segmentation and tracking.

**Human Segmentation and Tracking.** For segmentation, we predict whether each patch $(i, j)$ contains human parts by generating a score vector $\mathbf{m}^{i,j} \in \mathbb{R}^{(16 \times 16) \times 1}$ from the corresponding image token $\mathbf{F}^{i,j} \in \mathbb{R}^c$. This is achieved by passing $\mathbf{F}^{i,j}$ through an MLP, applying a sigmoid activation, and then using pixel shuffle [81] to produce a pixel-aligned

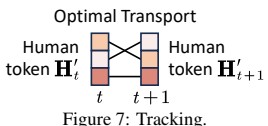

Optimal Transport

Figure 7: Tracking.

dense mask: $\mathbf{m}^{i,j} = \texttt{PixelShuffle}\left(\sigma\left(\mathrm{MLP}_{\mathrm{mask}}(\mathbf{F}^{i,j})\right)\right)$. We perform human tracking by leveraging the discriminative features encoded in the refined human token $\mathbf{H}'$, which encapsulates both human identity and human parameters. This enables us to formulate human tracking as a feature matching problem [77], where tracklet association is achieved by matching the refined tokens across timesteps. We maintain a human token tracklet [69] indexed by $\mathcal{A} = \{1, \ldots, M\}$ after each step of the online processing, which allows us to build a memory bank for all observed humans, and derive soft assignment matrix $\mathbf{A} \in [0, 1]^{M \times N}$ for current detections indexed by $\mathcal{B} = \{1, \ldots, N\}$. To estimate the likelihood of a given tracklet-detection pair, we use the pairwise L2 distance $\mathbf{D}_{m,n} = ||\mathbf{H}^{m'} - \mathbf{H}^{n'}||_2, \ \forall (m, n) \in \mathcal{A} \times \mathcal{B}$ to obtain the cost matrix $\mathbf{D} \in \mathbb{R}^{M \times N}$. To suppress unmatched human tokens, we augment the cost $\mathbf{D}$ to $\overline{\mathbf{D}} \in \mathbb{R}^{(M+1) \times (N+1)}$ by appending a new row and column dustbin with a threshold, so that unmatched human tokens are explicitly assigned to it. The assignment with dustbin $\overline{\mathbf{A}}$ can be solved by optimal transport [67] with the Sinkhorn algorithm [20] to minimize the total cost $\sum_{m,n} \overline{\mathbf{D}}_{m,n} \overline{\mathbf{A}}_{m,n}$ under the constraints of $\overline{\mathbf{A}} \, \mathbf{1}_{N+1} = \mathbf{a}$ and $\overline{\mathbf{A}}^\top \mathbf{1}_{M+1} = \mathbf{b}$, where $\mathbf{a} = [\mathbf{1}_M^\top \ N]^\top$ and $\mathbf{b} = [\mathbf{1}_N^\top \ M]^\top$, denote the number of expected matches for each human token and dustbin in $\mathcal{A}$ and $\mathcal{B}$.

**Training Strategy.** We finetune CUT3R on a synthetic dataset, BEDLAM [5], which is small-scale yet high-quality, with 6k sequences featuring 3D scene depth, camera poses, and SMPL-X meshes [64] of multiple persons in the world coordinates. Following CUT3R and MASt3R, we apply a confidence-aware 3D regression loss $\mathcal{L}_{\mathrm{pointmap}}$ to the metric-scale pointmaps, as well as a camera pose loss $\mathcal{L}_{\mathrm{pose}}$ to the ground-truth camera poses. This helps prevent CUT3R from forgetting the rich spatial and temporal priors learned from large-scale 3D scene datasets. To readout human from CUT3R, we follow Multi-HMR to minimize a binary cross-entropy loss $\mathcal{L}_{\mathrm{detection}}$ on $s^{i,j}$, L1 regression losses $\mathcal{L}_{\mathrm{smpl}}$ to human parameter $\mathbf{Y}_t$, $\mathcal{L}_{\mathrm{mesh}}$ to explicit human meshes, and reprojection loss $\mathcal{L}_{\mathrm{reproj.}}$. With our efficient *human prompt tuning* protocol, Human3R requires just one day of training on a single NVIDIA 48GB GPU, and still achieves state-of-the-art performance. Please checkout more training details in Appendix C of *Sup.Mat.*

**Test-Time Sequence Length Adaptation.** Trained with sequences of only 4 images, we observe that performance of Human3R degrades when the inference sequence length exceeds the training context. This is a common issue for RNN-based methods [9, 14, 95, 104], including CUT3R [97], where the state tends to forget earlier frames, resulting in significant performance drops as the number of input views increases. To address this limitation and support longer sequence, we adopt TTT3R [12], which parameterizes the state $\mathbf{S}$ as a fast weight [78] and updates it using gradient descent: $\mathbf{S}_t = \mathbf{S}_{t-1} - \beta_t \nabla(\mathbf{S}_{t-1}, \mathbf{F}_t, \mathbf{z})$, where $\nabla(\mathbf{S}_{t-1}, \mathbf{F}_t, \mathbf{z})$ denotes the gradient function and $\beta_t$ is the learning rate. Intuitively, this Test-Time Training (TTT) [90] procedure adaptively encodes the current observation into the memory state using a dynamic learning rate, enabling online adaptation. This approach effectively balances the retention of historical context with the integration of new observations. We follow TTT3R to use the spatial average of the attention values as a closed-form update rule for online associative recall in test time, and formulate the state update as: $\mathbf{S}_t = \mathbf{S}_{t-1} - \beta_t \nabla(\mathbf{S}_{t-1}, \mathbf{F}_t, \mathbf{z}, \mathbf{H}_t)$. Inspired by the correlation between length generalization and unexplored state distributions [75], we further propose a state reset process: the state is reset every 100 frames, using the global camera pose as a cue to align the resulting chunks.

# 4 EXPERIMENTS

We unfold the validation of Human3R and the baselines on human mesh recovery in the camera coordinates (Sec. 4.1) and the world coordinates (Sec. 4.2) respectively, and then compare our model with current state-of-the-art genetic 3D reconstruction methods in camera pose estimation and video depth estimation (Sec. 4.3). We also analyze the components of Human3R in Sec. 4.5.

| Category | Method | Crop-free | Detection-free | Intrinsic-free | 3DPW (14) | | | EMDB-1 (24) | | |
|---|---|---|---|---|---|---|---|---|---|---|
| | | | | | PA-MPJPE ↓ | MPJPE ↓ | PVE ↓ | PA-MPJPE ↓ | MPJPE ↓ | PVE ↓ |
| Multi-stage | CLIFF [52] | ✗ | ✗ | ✗ | 43.0 | 69.0 | 81.2 | 68.3 | 103.3 | 123.7 |
| | HMR2.0a [31] | ✗ | ✗ | ✓ | 44.4 | 69.8 | 82.2 | 61.5 | 97.8 | 120.0 |
| | TokenHMR [25] | ✗ | ✗ | ✓ | 44.3 | 71.0 | 84.6 | 55.6 | 91.7 | 109.4 |
| | CameraHMR [62] | ✗ | ✗ | ✓ | 38.5 | 62.1 | 72.9 | 43.7 | 73.0 | 85.4 |
| | NLF [76] | ✗ | ✗ | ✗ | 37.3 | 60.3 | 71.4 | 41.2 | 69.6 | 82.4 |
| | PromptHMR [100] | ✓ | ✗ | ✗ | 36.6 | 58.7 | 69.4 | 41.0 | 71.7 | 84.5 |
| One-stage | BEV [88] | ✓ | ✓ | ✓ | 46.9 | 78.5 | 92.3 | 70.9 | 112.2 | 133.4 |
| | Multi-HMR [3] | ✓ | ✓ | ✗ | 45.9 | 73.1 | 87.1 | 50.1 | 81.6 | 95.7 |
| | **Human3R** | ✓ | ✓ | ✓ | 44.1 | 71.2 | 84.9 | 48.5 | 73.9 | 86.0 |

Table 1: **Evaluation of local human mesh reconstruction** on 3DPW [94] and EMDB-1 [39] datasets.

| Category | Method | Preprocessed Input (✓ = not required) | | | | | | | Output | | | EMDB-2 (24) | | | RICH (24) | | |
|---|---|---|---|---|---|---|---|---|---|---|---|---|---|---|---|---|---|
| | | Detection | Tracking | LocalHuman | Camera | Mask | Depth | Contact | GlobalHuman | CameraPose | Scene | WA-MPJPE ↓ | W-MPJPE ↓ | RTE ↓ | WA-MPJPE ↓ | W-MPJPE ↓ | RTE ↓ |
| Offline | GLAMR [110] | ✗ | ✗ | ✗ | ✓ | ✗ | ✗ | ✓ | ✓ | ✗ | ✗ | 280.8 | 726.6 | 11.4 | 129.4 | 236.2 | 3.8 |
| | SLAHMR [109] | ✗ | ✗ | ✗ | ✗ | ✓ | ✓ | ✓ | ✓ | ✗ | ✗ | 326.9 | 776.1 | 10.2 | 132.2 | 237.1 | 6.4 |
| | COIN [48] | ✗ | ✗ | ✗ | ✗ | ✓ | ✓ | ✓ | ✓ | ✗ | ✗ | 152.8 | 407.3 | 3.5 | 169.5 | 254.5 | - |
| | GVHMR [80] | ✗ | ✗ | ✗ | ✗ | ✓ | ✓ | ✓ | ✓ | ✗ | ✗ | 111.0 | 276.5 | 2.0 | 78.8 | 126.3 | 2.4 |
| | TRAM [99] | ✗ | ✗ | ✗ | ✗ | ✗ | ✗ | ✓ | ✓ | ✓ | ✗ | 76.4 | 222.4 | 1.4 | 127.8 | 238.0 | 6.0 |
| | JOSH [53] | ✗ | ✗ | ✗ | ✗ | ✗ | ✗ | ✗ | ✓ | ✓ | ✓ | 68.9 | 174.7 | 1.3 | 89.0 | 132.5 | 3.0 |
| Online | TRACE [89] | ✓ | ✓ | ✓ | ✓ | ✓ | ✓ | ✓ | ✓ | ✗ | ✗ | 529.0 | 1702.3 | 17.7 | 238.1 | 925.4 | 610.4 |
| | WHAM [82] | ✗ | ✗ | ✗ | ✗ | ✓ | ✓ | ✓ | ✓ | ✗ | ✗ | 135.6 | 354.8 | 6.0 | 108.4 | 196.1 | 4.5 |
| | JOSH3R [53] | ✗ | ✗ | ✗ | ✓ | ✗ | ✓ | ✓ | ✓ | ✓ | ✓ | 220.0 | 661.7 | 13.1 | - | - | - |
| | **Human3R** | ✓ | ✓ | ✓ | ✓ | ✓ | ✓ | ✓ | ✓ | ✓ | ✓ | 112.2 | 267.9 | 2.2 | 110.0 | 184.9 | 3.3 |

Table 2: **Evaluation of global human motion estimation** on EMDB-2 [39] and RICH [33] datasets.

## 4.1 LOCAL HUMAN MESH RECONSTRUCTION

We evaluate human pose and shape reconstruction in camera coordinates on 3DPW [94] and EMDB (subset 1) [39], and follow the commonly used local human mesh reconstruction metrics as prior works [3, 100]: mean per-joint position error (MPJPE), Procrustes-aligned per-joint position error (PA-MPJPE), and per-vertex error (PVE) measured in millimeters ($mm$).

We compare with both multi-stage and one-stage leading methods in Tab. 1. Most multi-stage methods rely on human detection and cropping, processing each detected person individually. Without additional cropping, PromptHMR [100] takes the full image as input and prompt it with bounding-box prompts, and achieves strong performance. Among one-stage models, Multi-HMR [3] eliminates the need for off-the-shelf human detectors, but still requires ground-truth camera intrinsics. BEV [88] removes the dependency on ground-truth intrinsics, aligning with our experimental setting. Our approach surpasses these methods across all metrics, demonstrating substantial performance improvements (10% improvement on MPJPE and PVE on EMDB-1), which we attribute to the spatiotemporal awareness provided by CUT3R as a generic 4D reconstruction model.

## 4.2 GLOBAL HUMAN MOTION ESTIMATION

We evaluate motion and trajectory estimation accuracy in world coordinates on EMDB (subset 2) [39] and RICH [33], both feature long sequences with ground-truth global human trajectories and meshes. Following previous work [82, 99], we divide each sequence into 100-frame segments and evaluate 3D joint errors using two metrics: W-MPJPE, which aligns the first two frames, and WA-MPJPE, which aligns the entire segment. Both metrics are reported in millimeters ($mm$). To comprehensively assess trajectory accuracy over long sequences, we additionally report the root translation error (RTE, in %) after rigid alignment (without scaling), normalized by the total displacement.

We compare with both offline and online methods in Tab. 2. Given multiple offline pre-cached conditions, GVHMR [80] and JOSH [53] respectively achieve strong performance on sequences with static cameras (RICH) and long human trajectories (EMDB-2). JOSH3R [53], trained with multi-stage pseudo ground truth from JOSH, removes the need for pre-cached camera poses, depth, contact, and iterative refinement. It enables online prediction of global human trajectories, scene points, and camera poses, but with a $2\times$ drop in accuracy compared to WHAM and still requires precomputed human detection, segmentation, and meshes in camera coordinates. TRACE [89] takes only RGB video as input, matching our experimental setting, but outputs only global human meshes. In contrast, our method also reconstructs scene geometry and estimates camera poses. In summary, Human3R jointly reconstructs multiple human meshes and trajectories in world space, scene geometry, and camera poses, achieving notable gains (20% lower W-MPJPE and 60% lower RTE against WHAM on EMDB-2), while enabling online inference and end-to-end training. We visualize the global human motion estimation within the dense scene, together with the predicted camera trajectory, in Fig. 8.

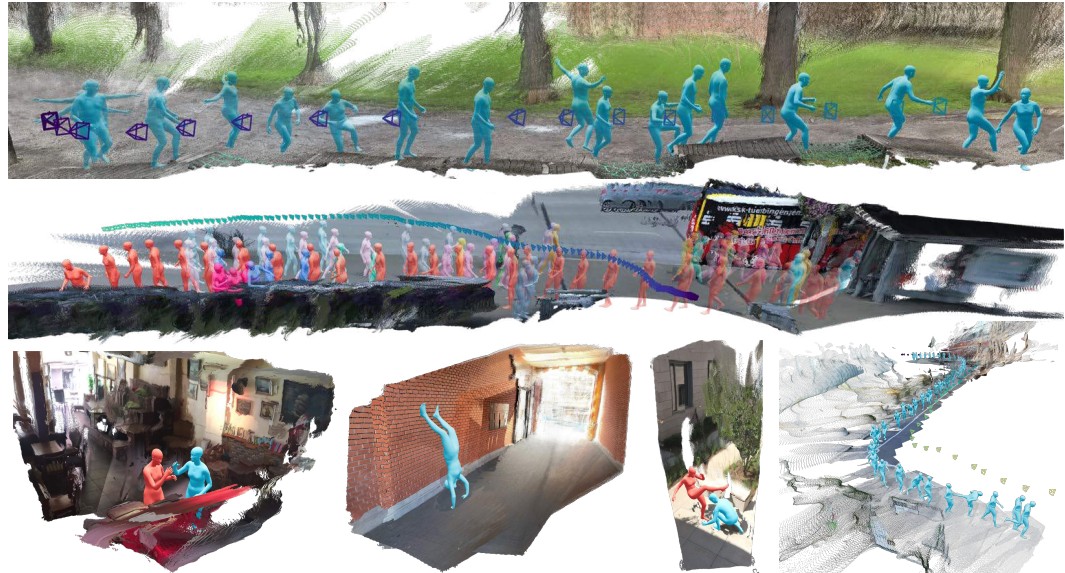

Figure 8: **Qualitative 4D human-scene reconstruction results.** Given video captured from a single camera, Human3R performs online reasoning about global human motion, the surrounding environment, and camera poses *all at once*. ➤ Check our `website` for video results.

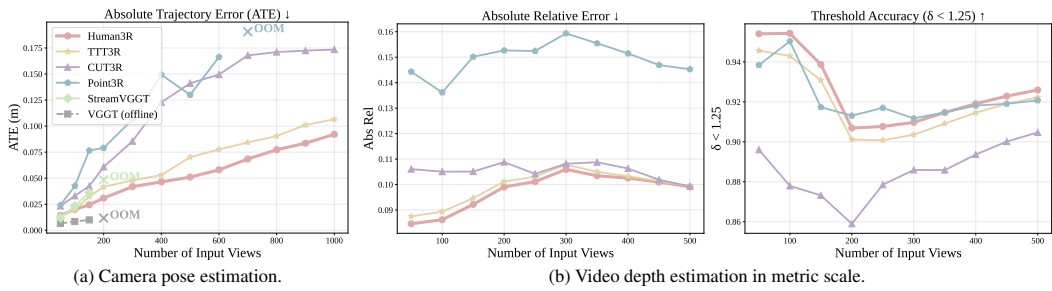

(a) Camera pose estimation.      (b) Video depth estimation in metric scale.

Figure 9: **Evaluation of generic 3D reconstruction** with camera pose estimation on TUM-D [86] and video depth estimation on Bonn [61].

## 4.3 GENERIC 3D RECONSTRUCTION

**Camera Pose Estimation.** Following prior works [12, 97], we evaluate camera pose estimation accuracy on TUM dynamics [86] dataset with dynamic humans. We report the Absolute Translation Error (ATE) after applying the Sim(3) alignment [92] on the estimated camera trajectory to the ground-truth. We compare with current leading 3D reconstruction foundation models [12, 96, 97, 104, 117].

We include VGGT, an offline method utilizing full attention, as an upper bound for online approaches, since it retains complete historical context without forgetting. VGGT and StreamVGGT rely on full attention, making them relatively slow and prone to running out of memory (OOM). In contrast, CUT3R maintains consistently low GPU usage and enables online inference, but struggles to remember long sequences, resulting in less accurate pose estimation. TTT3R [12] introduces a closed-form state transition rule as a training-free intervention to mitigate the catastrophic forgetting observed in CUT3R. As shown in Fig. 9a, integrating TTT3R with Human3R leads to further improvements in camera pose estimation after *human prompt tuning* compared to the original TTT3R.

**Video Depth Estimation.** Following common practice [12, 97], we evaluate video depth estimation on Bonn [61] datasets with dynamic humans. We use Absolute Relative Error and $\delta < 1.25$ (percentage of predicted depths within a 1.25-factor of true depth) as metrics. Metric scale video depth estimation evaluates per-frame depth quality and inter-frame depth consistency without per-sequence scale or shift alignment, which measures the absolute depth accuracy. Fig. 9b presents the quantitative comparison between our method and the online baselines, and still Human3R+TTT3R achieves more acccurate depth estimation over naive TTT3R. We do not plot VGGT [96] and StreamVGGT [117] for the evaluation of the metric depth, as they can only predict the relative depth without metric scale.

By integrating TTT3R and fine-tuning with *human prompt tuning* on human-scene 4D datasets, our approach achieves SOTA human mesh recovery and also slightly improves generic 3D reconstruction. This highlights the mutual benefits of jointly reasoning about humans and scenes.

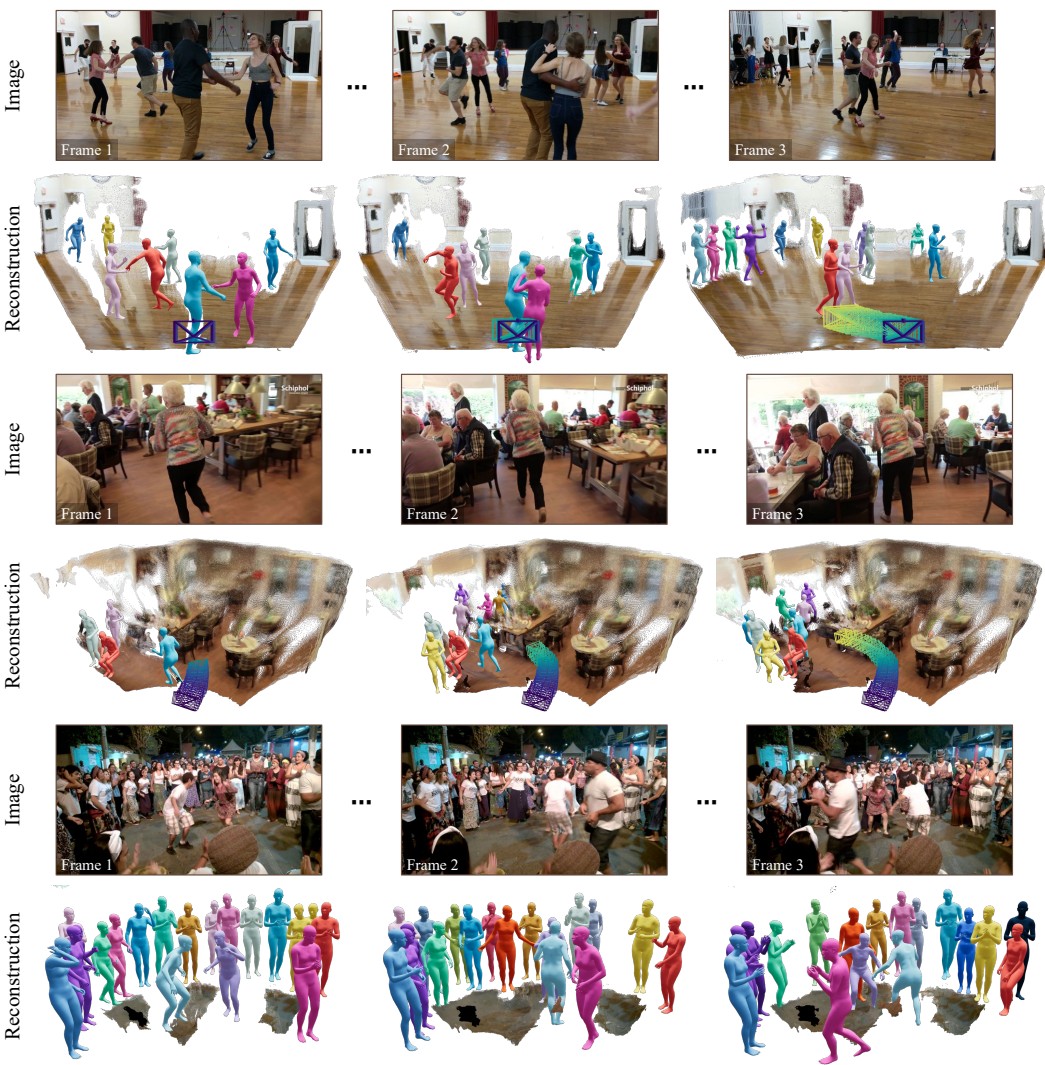

Figure 10: **Qualitative Results on Crowded Scenes.** Given a dynamic multi-person stream, Human3R incrementally reconstructs the 3D scene and humans in the global frame. We visualize the accumulated scene and camera trajectory, with current-frame human meshes colored by tracking IDs. Notably, our model generalizes to in-the-wild crowds (>10 people) despite being trained on synthetic data with only 1-10 subjects, operating in an efficient one-shot scheme without external offline modules.

## 4.4 GENERALIZATION TO CROWDED SCENES

Current quantitative evaluations do not reflect performance in crowded scenarios, as ground-truth benchmarks typically contain only one (EMDB [39], RICH [33]) or two people (3DPW [94]). Therefore, we conduct qualitative evaluations on out-of-distribution multi-person sequences to assess the accuracy of both human and scene reconstruction. Figure 10 demonstrates that Human3R robustly estimates 3D scene structures and camera poses even when the view is heavily occluded by humans. Simultaneously, the observed humans are reconstructed accurately within the global 3D scene, exhibiting consistent motion trajectories and stable ID tracking.

Crucially, Human3R operates in a single feed-forward pass. Unlike top-down methods, which have inference times that increase linearly with the number of people (due to per-person HMR), we follow the bottom-up Multi-HMR [3] to recover multiple human meshes in a one-shot scheme, extending this capability to reconstruct humans in the world frame jointly with the 3D scene. This ensures that the inference speed remains constant regardless of crowd density.

However, we observe limitations when multiple humans are heavily occluded to the extent that they occupy the same head token. In such cases, Human3R struggles to differentiate subjects, as our method relies on the head token as the primary discriminative query prompt. We provide further analysis of robustness under severe occlusion in Appendix A.1 of *Sup.Mat.*

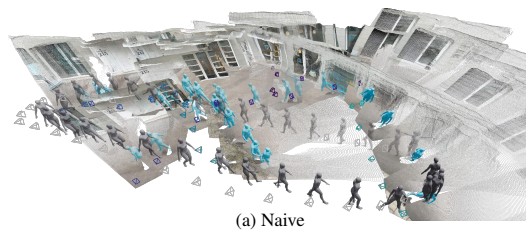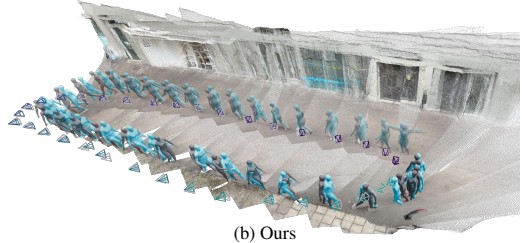

|  (a) Naive  |  (b) Ours  |

Figure 11: **Comparison with naive CUT3R+Multi-HMR combination** in global human motion, 3D scene reconstruction, and camera poses estimation. The colors ● and ● indicates Prediction and Ground-truth, respectively. 🔍 See Fig. 19 in *Sup.Mat.* for a zoomed-in visualization.

## 4.5 ANALYSIS

**1) Human3R benefits from the 3D awareness of CUT3R**. We use the Mean Root Position Error (MRPE) [3] between the predicted and ground-truth pelvis locations to evaluate the quality of spatial location estimation. As shown in Fig. 12, Multi-HMR performance varies when processing images at different aspect ratios, while Human3R performs consistently well without requiring camera intrinsics. The metric-scale 3D scene context guides multi-human recovery by capturing their relative spatial relationships, thereby improving our intrinsic robustness. This enables Human3R to recover coherent 3D humans from intrinsic-agnostic internet images. See more in-the-wild examples on our `website`.

**2) Human3R benefits from the human awareness of Multi-HMR**. To enhance the details of reconstructed human pose and shape, we introduce Multi-HMR [3] ViT DINO encoder that fine-tuned on human-specific datasets as human prior. As shown in Tab. 3, Human3R reconstructs more fine-grained human pose and shape when injecting human priors in better detail.

**3) Human3R takes the best of both worlds**. Human3R predicts better camera poses and scenes than CUT3R (Fig. 9), better local humans than Multi-HMR (Tab. 1 and Fig. 12), and better global humans than the naive combinations of Multi-HMR and CUT3R (Tab. 3), *all-at-once*.

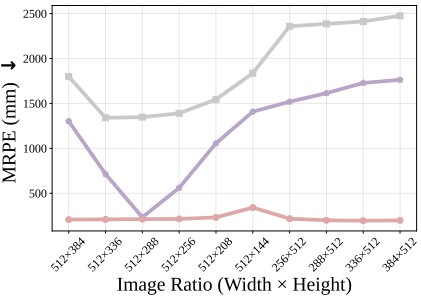

Figure 12: **Evaluation of intrinsic robustness**. Multi-HMR w/ GT intrinsics and Multi-HMR w/o GT intrinsics are sensitive to image aspect ratios, Human3R performs consistently well without camera intrinsics.

| Ablations | WA-MPJPE ↓ | W-MPJPE ↓ | RTE ↓ |
|---|---|---|---|
| Human3R w/o Prior | 221.2 | 808.4 | **2.2** |
| Human3R w/ ViT-S/672 | 129.9 | 314.2 | **2.2** |
| Human3R w/ ViT-B/672 | 122.1 | 292.9 | **2.2** |
| Human3R w/ ViT-L/672 | 113.6 | 291.7 | **2.2** |
| Human3R w/ ViT-L/896 | **112.2** | **267.9** | **2.2** |
| Naive w/o TTT3R | 455.4 | 1263 | 14.3 |
| Naive w/ TTT3R | 401.3 | 1173.9 | 12.2 |
| Human3R w/o TTT3R | 124.3 | 292.3 | 2.5 |
| Human3R w/ TTT3R | **112.2** | **267.9** | **2.2** |

Table 3: **Ablation of human prior and naive baselines** in global human motion on EMDB-2 dataset, using different Multi-HMR ViT-DINO encoders and a simple combination of Multi-HMR and CUT3R as the naive baseline. 🔍 Please check more detailed analyses in Appendix A.6, Appendix A.5, and Appendix A.7 of *Sup.Mat.*

## 5 CONCLUSION

We presented Human3R, a one-stage method for 4D human-scene reconstruction, providing a feasible strategy for both efficient finetuning and real-time inference. Our method demonstrates competitive or state-of-the-art performance in both human motion recovery and general 3D reconstruction benchmark, and generalizes to casually captured videos.

**Limitations & Future Work.** Human3R represents an important first step towards feed-forward 3D human and scene reconstruction, but several limitations remain. First, our method relies on the head as the discriminative keypoint for detecting humans, which leads to failures when the head is not visible. Incorporating pixel-aligned body point localizers [40, 76] could mitigate this issue. Second, we currently represent humans using proxy SMPL meshes that do not model clothing or appearance. Extending the framework with 3DGS anchored on SMPL would enable richer, more holistic reconstructions. Third, while Human3R is designed as an online method for real-time applications, it can also serve as an effective initialization for optimization-based approaches [53] to improve accuracy at the cost of additional computation. Beyond these limitations, Human3R opens avenues for broader applications. Although our focus is on reconstructing humans from monocular videos, the underlying principles can extend to other dynamic entities. By leveraging spatial and temporal cues, the framework could be adapted to reconstruct animals, vehicles, or other moving objects with full 6D poses (see limitations Appendix D of *Sup.Mat.*). Such extensions would enable applications in wildlife monitoring, traffic analysis, human-object interaction, and robotics.

## 6 ACKNOWLEDGE

Thank all members of Endless AI, Inception3D and RVH Group for help, and Yiru for creating the fantastic logo — love it! Yue and Xingyu are funded by the Westlake Education Foundation. Gerard and Yuxuan are funded by the Carl Zeiss Foundation, the Deutsche Forschungsgemeinschaft - 409792180 (EmmyNoether Programme, project: Real Virtual Humans), and the German Federal Ministry of Education and Research: Tübingen AI Center, FKZ: 01IS18039A. Gerard is a member of the Machine Learning Cluster of Excellence, EXC number 2064/1 – Project number 390727645.

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

# Appendix

## Table of Contents

## A  ANALYSIS

### A.1  ROBUSTNESS TO TRUNCATION AND OCCLUSION

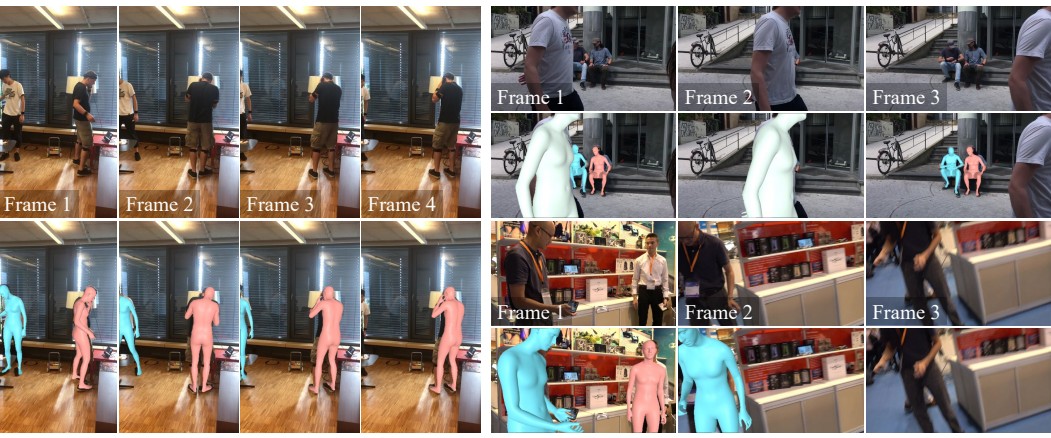

Figure 13: **Truncation Examples.** We demonstrate the transition from successful human detection to a miss as the head becomes increasingly truncated. Human3R exhibits robust performance, successfully detecting humans when (**Left**) only a minimal portion of the head is visible, or (**Right**) detection relies solely on anatomical points adjacent to the head (e.g., the chest, back, or neck).

We observe that Human3R sometimes struggles to detect humans when the head is not visible due to severe truncation or occlusion, as our method relies on the head as the primary discriminative keypoint. However, Human3R demonstrates significant robustness: it remains capable of detecting humans when only a small portion of the head is visible (as shown in Fig. 13 left and Fig. 14) or when anatomical points adjacent to the head (e.g., chest, back, or neck) are visible (as shown in Fig. 13 right and Fig. 14).

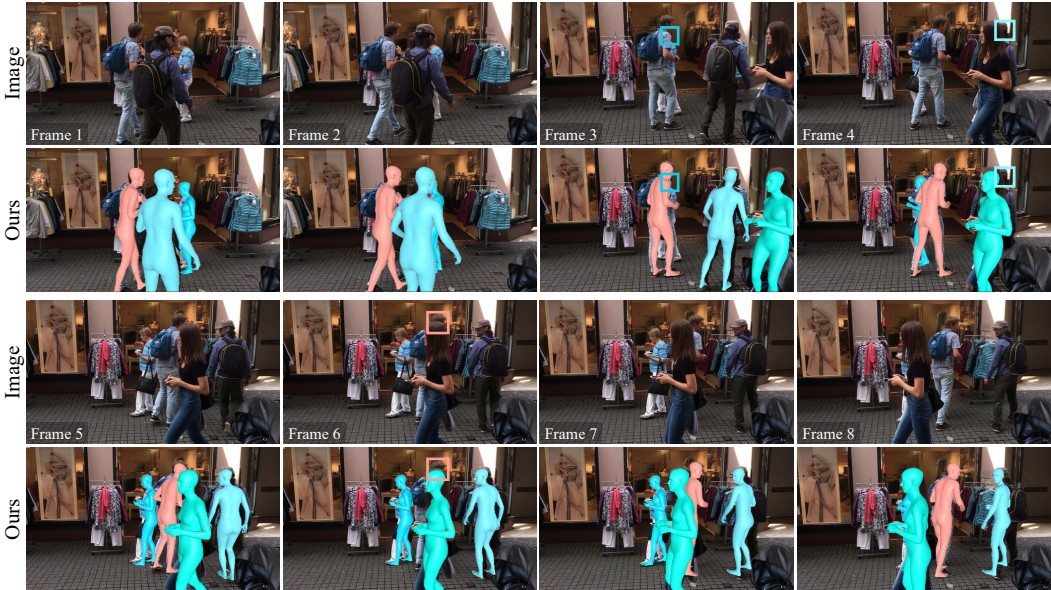

Figure 14: **Occlusion Examples.** In crowded scenarios, head detection misses may occur when multiple individuals occupy the same head token (highlighted boxes). However, this does not lead to identity switches or lost tracks (colored meshes); our method successfully re-associates the subject and continues tracking once the head or adjacent parts are visible.

The model's general robustness to occlusion stems from our training strategy, which follows the data augmentation protocol from Multi-HMR [3]. Specifically, we apply image cropping during training. If the head is truncated, we supervise the network to identify the visible point closest to the unobserved head center, such as the chest or neck. We hypothesize that incorporating more aggressive cropping augmentations and training on heavily occluded datasets would further enhance robustness and reduce misses.

To eliminate the single-point dependency, future work could employ pixel-aligned body point localizers [40, 76] to localize multiple canonical body points in pixel space as prompts, rather than relying solely on the head. This could be followed by regressing the corresponding SMPL parameters and a post-processing step to merge multiple queries. Another promising avenue, following PromptHMR [100], is to employ bounding boxes or segmentation maps as prompts. While these also operate within the pixel space, they benefit from being region-based, allowing the model to identify humans based on any observable body region rather than specific keypoints.

## A.2 ROBUSTNESS TO NON-FULL BODY CAPTURES

Standard quantitative evaluations focus primarily on full-body sequences, as most ground-truth benchmarks lack non-full body data. Consequently, we conduct qualitative evaluations on challenging quarter/upper-body views (Fig. 15), face-focused close-ups (Fig. 16, left and middle), and out-of-distribution (OOD) human poses (Fig. 16, right).

As shown in Fig. 15, Human3R reconstructs consistent 3D scenes and camera poses even when the image is mostly occupied by the upper-body. Leveraging a strong learned prior of scene-aware human poses, our model not only reconstructs the observable upper body but also infers the unobserved lower body, yielding physically plausible poses that stand on the reconstructed ground plane.

However, humans also perform dynamic movements like running rather than merely standing. In face-focused running sequences (e.g., live broadcasts, Fig. 16 left), Human3R correctly reconstructs the scene and camera but fails to generate the running motion, defaulting to a standing pose. This reveals a limitation of our deterministic pipeline, which tends to regress the average pose when visual evidence is missing. Integrating generative modeling to capture multimodal distributions represents a promising direction for future work.

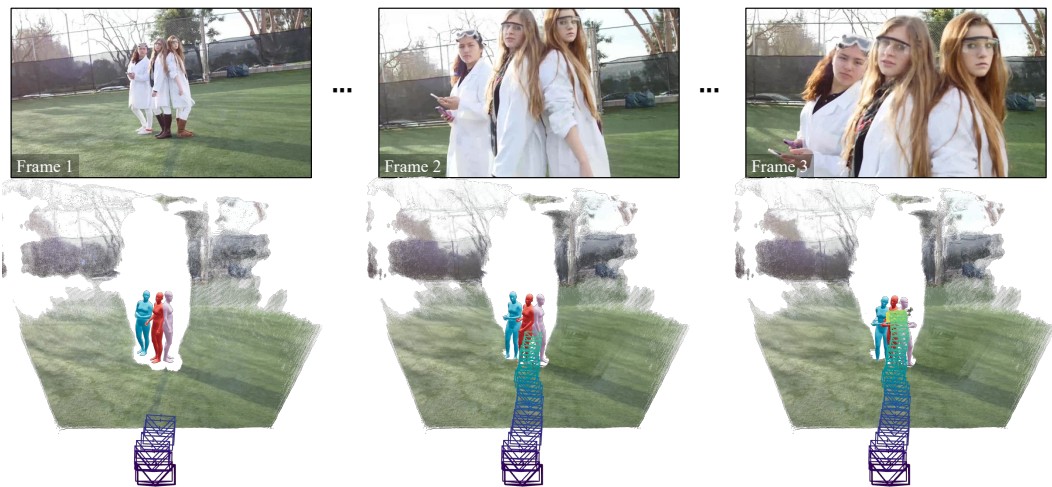

Figure 15: **Challenging Quarter/upper-body Examples.** We visualize the accumulated 3D scene, camera trajectory, and current-frame human meshes from a video captured with dolly-in camera motions. Guided by a scene-aware human pose prior, our model infers physically plausible lower-body poses that maintain ground contact.

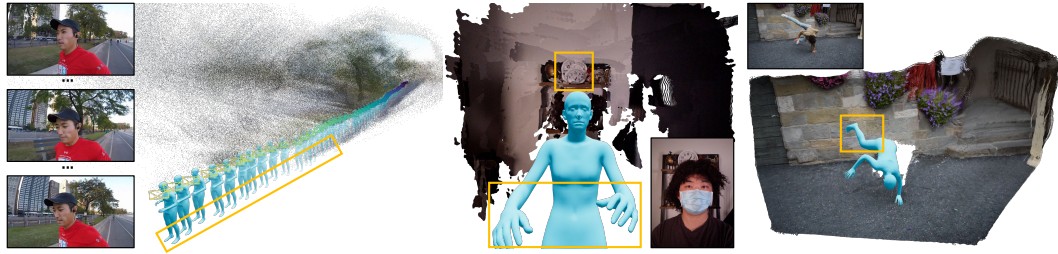

Figure 16: **Limitations on OOD scenarios. Left:** Our deterministic pipeline fails to infer the running poses of the lower body in partial views. **Middle:** Close-up scenarios (e.g., selfies) exhibit misalignment of the wall clock and hallucination of unobserved hands. **Right:** Extreme poses (e.g., breakdancing) lead to performance degradation.

Furthermore, we observe performance degradation in fine-grained scene reconstruction (e.g., the misaligned wall clock, Fig. 16 middle), unobserved extremities reasoning (e.g., hands in selfies, Fig. 16 middle), and extreme OOD human poses (e.g., breakdancing, Fig. 16 right). Addressing these limitations may require scaling up the training dataset and exploring self-supervised solutions.

## A.3 COMPARISON OF HUMAN MESH PROJECTION

We further evaluate the quality of our pixel-aligned predictions and mesh-camera consistency by projecting the reconstructed mesh back onto the image plane for overlay and comparison against baseline methods. As illustrated in Fig. 17, we visualize the human mesh projection and compare it with state-of-the-art global human mesh recovery (HMR) methods, GVHMR [80] and TRAM [99], using long sequences from the 3DPW dataset. GVHMR produces superior human meshes that project to fit the subject's body contour well. However, it is limited to estimating a single person, typically the one occupying the largest number of pixels, and thus often fails to handle videos with complex multi-person occlusion. TRAM, while exhibiting slightly less accurate pixel-aligned mesh projection than GVHMR, natively models multi-person human mesh recovery, leading to more robust and expressive results in crowded scenes. Our method combines the strengths of both approaches: it achieves mesh projection accuracy comparable to GVHMR while simultaneously retaining the robust multi-person human mesh recovery capability as TRAM (also shown in Sec. 4.4).

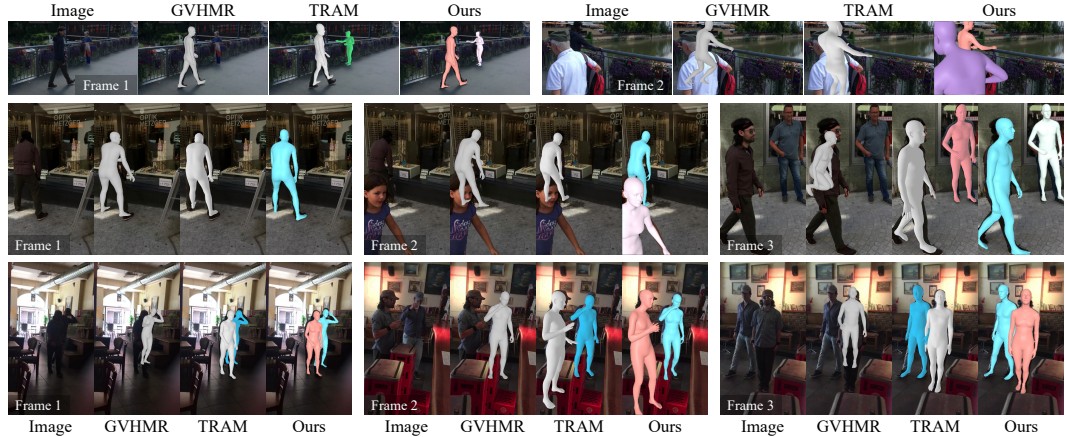

Figure 17: **Mesh Projection Visualization.** We compare our method with GVHMR [80] and TRAM [99] on long sequences of the 3DPW [94] dataset. GVHMR yields the best fit for human contours but is limited to single-person reconstruction, struggling with multi-person occlusion. TRAM can handle multi-human mesh recovery, albeit with slightly inferior projection accuracy. Our method demonstrates robustness in crowed scenarios with severe occlusion while maintaining comparable pixel-aligned mesh accuracy.

| Models | Runtime (FPS) on a NVIDIA RTX4090 | | | | | |
|---|---|---|---|---|---|---|
| | 3DPW (288×512) | BEDLAM (512×288) | RICH (512×368) | EMDB (384×512) | Bonn (512×384) | TUM-D (512×384) |
| Human3R w/ ViT-S/672 | 15.87 | 15.64 | 14.28 | 13.75 | 13.65 | 13.59 |
| Human3R w/ ViT-B/672 | 13.33 | 12.69 | 11.89 | 11.68 | 11.67 | 12.41 |
| Human3R w/ ViT-L/672 | 9.17 | 8.73 | 8.38 | 8.27 | 8.27 | 8.61 |
| Human3R w/ ViT-L/896 | 5.38 | 5.3 | 5.15 | 5.09 | 5.06 | 5.15 |

Table 4: **Inference speed** of Human3R variants using different ViT backbones on six benchmark datasets, spanning scenarios from synthetic to real-world, single-person to multi-person, indoor to outdoor, and short to long sequences. The lightweight Human3R (ViT-S/672) operates at real-time speed, and even our largest model (Human3R w/ ViT-L/896) maintains a competitive inference speed (> 5 FPS) across all datasets. Results are obtained on an NVIDIA RTX 4090 GPU with dual Intel Xeon Gold 6530 CPUs.

## A.4 COMPUTATION EFFICIENCY

To comprehensively evaluate the efficiency of our method, we conduct runtime (FPS, frames per second) experiments on an NVIDIA RTX 4090 GPU with dual Intel Xeon Gold 6530 processors (total 64 physical cores, 128 logical threads). As reported in Tab. 4, we benchmark diverse backbones (ViT-S/672, ViT-B/672, ViT-L/672, and ViT-L/896) across multiple datasets with varying input resolutions: 3DPW [94] (288×512), BEDLAM [5] (512×288), RICH [33] (512×368), EMDB [39] (384×512), Bonn [61] (512×384), and TUM-D [86] (512×384).

We observe that Human3R exhibits consistent runtime performance across different datasets and image resolutions. Specifically, the lightweight Human3R (ViT-S/672) operates at real-time speeds, ranging from 13.59 to 15.87 FPS. Although increasing the model size naturally incurs higher computational costs, our largest model (Human3R w/ ViT-L/896) still maintains a competitive speed of 5.06 – 5.38 FPS. We further analyze the trade-off between inference speed and reconstruction accuracy in Tab. 8 of Appendix A.5.

We compare the computational efficiency of Human3R against state-of-the-art global human mesh recovery (HMR) methods: GVHMR [80] and TRAM [99]. It is important to note that unlike our method, these baselines are limited to human reconstruction and do not recover the 3D scene. Regarding JOSH [53], which jointly reconstructs humans and scenes, we exclude a detailed breakdown as the code is not yet public. Their reported overall runtime on an NVIDIA RTX 4090 GPU is approximately 0.8 FPS.

As detailed in Tab. 5, the GVHMR pipeline relies on extensive preprocessing to extract four feature types: bounding box detection (YOLOv8), 2D keypoints (ViTPose), image features (ViT), and camera poses (DPVO). Individual MLPs map these features to a common dimension, followed by element-wise summation to generate per-frame tokens for the HMR network. Notably, these costs apply to single-person reconstruction only, as GVHMR does not support multi-person scenarios. The complete offline process operates at 4.88 FPS.

| Methods | GVHMR (Single-human) |
|---|---|
| YOLOv8 (Detection) | 0.041 |
| ViTPose (2D Keypoint) | 0.067 |
| ViT Feature Extraction | 0.059 |
| DPVO (SLAM) | 0.037 |
| HMR | 0.001 |
| **Total (s)** | 0.205 |
| **FPS** | 4.88 |

Table 5: **Runtime analysis of GVHMR.** We report the model inference time (average runtime per frame in seconds). Analysis are tested on the 3DPW sequences.

The latency breakdown for TRAM is shown in Tab. 6. This method requires computationally intensive preprocessing, including detection and tracking (DEVA), segmentation (SAM), camera pose estimation (DROID-SLAM), and metric depth estimation (ZoeDepth). These priors are integrated via global alignment, followed by an HMR network for local human reconstruction. Finally, all camera-space estimations are transformed into the world coordinate using the globally aligned camera poses. Consequently, although TRAM supports multi-person reconstruction, its pipeline is limited to offline operation at 0.86 FPS.

| Methods | TRAM (Multi-human) |
|---|---|
| DEVA (Detection&Track) | 0.3617 |
| SAM (Segmentation) | 0.1757 |
| DROID-SLAM | 0.0463 |
| ZoeDepth (Metric Depth) | 0.0120 |
| Global Alignment | 0.0049 |
| VIMO (HMR) | 0.5581 |
| **Total (s)** | 1.1587 |
| **FPS** | 0.86 |

Table 6: **Runtime analysis of TRAM.** We report the model inference time (average runtime per frame in seconds). Analysis are tested on the 3DPW sequences.

Dependencies associated with off-the-shelf estimators often impose synchronization barriers, which induce cumulative errors and hinder online inference. Furthermore, they face scalability bottlenecks like the *tabula rasa* problem [97]. To eliminate these dependencies, we design a unified framework for online inference, leveraging generalizable priors within an end-to-end, data-driven model.

Guided by the design principles of simplicity and scalability, Human3R adopts a streamlined architecture composed exclusively of an Encoder, a Decoder, and Heads. We also incorporate a Multi-HMR [3] ViT image encoder, initialized with pre-trained DINO [10, 60] weights and fully fine-tuned on human-specific datasets, to serve as a robust human prior. As detailed in Tab. 7, the computational cost distribution among the Encoder, Decoder, and Heads follows

| Modules | Human3R (Multi-human & Scene) | | | |
|---|---|---|---|---|
| | ViT-S/672 | ViT-B/672 | ViT-L/672 | ViT-L/896 |
| Encoder | 0.012 | 0.012 | 0.012 | 0.012 |
| Multi-HMR Encoder | 0.011 | 0.023 | 0.057 | 0.134 |
| Decoder | 0.019 | 0.019 | 0.019 | 0.019 |
| Heads | 0.021 | 0.021 | 0.021 | 0.021 |
| **Total (s)** | 0.063 | 0.075 | 0.109 | 0.186 |
| **FPS** | 15.87 | 13.33 | 9.17 | 5.38 |

Table 7: **Runtime analysis of Human3R components.** We report the inference time (average runtime per frame in seconds) of our model with different ViT backbones. Analysis are tested on the 3DPW sequences.

an approximate ratio of 1:2:2. In contrast, the latency of the Multi-HMR backbone varies significantly depending on model size (ranging from ViT-S/672 to ViT-L/896). As shown in Appendix A.5, the ViT-S/672 variant offers an optimal performance–speed trade-off, achieving approximately 15 FPS.

| | Human Mesh Reconstruction | | | | | | Global Human Motion | | | | | | |
|---|---|---|---|---|---|---|---|---|---|---|---|---|---|
| | 3DPW (14) | | | EMDB-1 (24) | | | EMDB-2 (24) | | | RICH (24) | | | |
| Ablations | PA-MPJPE ↓ | MPJPE ↓ | PVE ↓ | PA-MPJPE ↓ | MPJPE ↓ | PVE ↓ | WA-MPJPE ↓ | W-MPJPE ↓ | RTE ↓ | WA-MPJPE ↓ | W-MPJPE ↓ | RTE ↓ | FPS ↑ |
| Human3R w/o Prior | 102.1 | 173.5 | 200.4 | 145.8 | 214.0 | 252.7 | 221.2 | 808.4 | 2.2 | 226.0 | 399.5 | 3.4 | 18 |
| Human3R w/ ViT-S/672 | 56.1 | 87.8 | 103.1 | 66.9 | 93.6 | 106.5 | 129.9 | 314.2 | 2.2 | 131.8 | 208.3 | 3.3 | 15 |
| Human3R w/ ViT-B/672 | 49.3 | 79.6 | 94.3 | 56.6 | 84.1 | 96.4 | 122.1 | 292.9 | 2.2 | 119.2 | 188.3 | 3.3 | 11 |
| Human3R w/ ViT-L/672 | 48.5 | 83.1 | 96.7 | 54.1 | 82.9 | 95.0 | 113.6 | 291.7 | 2.2 | 110.3 | 185.0 | 3.3 | 7 |
| Human3R w/ ViT-L/896 | **44.1** | **71.2** | **84.9** | **48.5** | **73.9** | **86.0** | **112.2** | **267.9** | **2.2** | **110.0** | **184.9** | **3.3** | 5 |

Table 8: **Ablation of human prior** in human mesh reconstruction and global human motion estimation. To enhance the details of reconstructed human pose and shape, we introduce Multi-HMR [3] ViT DINO encoder that fine-tuned on human-specific datasets as human prior.

## A.5 HUMAN3R BENEFITS FROM THE HUMAN AWARENESS OF MULTI-HMR

To enhance fine-grained details in reconstructed human pose and shape, we incorporate the Multi-HMR [3] ViT DINO encoder fine-tuned on human-centric datasets, as a human prior. As shown in Tab. 8, injecting this prior enables Human3R to recover more detailed human pose and shape. We ablate Human3R without the prior (Human3R w/o Prior) and evaluate the impact of input image resolution (672 and 896) across Multi-HMR ViT DINO backbone sizes (ViT-S, ViT-B, ViT-L). Increasing the input resolution and model size consistently improves performance, at the cost of higher inference time, as reported on the right of Tab. 8 in frames per second (FPS). For global human motion estimation, a ViT-S backbone with $672 \times 672$ inputs offers a good accuracy-speed trade-off (approximately WA-MPJPE 100 and RTE 2 at 15 FPS). Higher resolutions and larger backbones can be more beneficial for detailed human-mesh reconstruction. This is expected, since fine details, such as facial expressions and hand poses, are better captured at higher resolution and by larger models with richer priors and higher-dimensional features. Without any extra compression or quantization efforts, the largest backbone (ViT-L) at $896 \times 896$ runs at 5 FPS while achieving accuracy competitive with multi-stage methods.

| Methods | In-distribution | | | | Out-of-distribution | | | |
|---|---|---|---|---|---|---|---|---|
| | PVE ↓ | PA-MPJPE ↓ | MPJPE ↓ | MRPE ↓ | PVE ↓ | PA-MPJPE ↓ | MPJPE ↓ | MRPE ↓ |
| Multi-HMR w/o GT K | 77.0 | 43.9 | 65.3 | 1347.5 | 140.1 | 69.8 | 116.0 | 2475.7 |
| Multi-HMR w/ GT K | 64.1 | 39.0 | 54.8 | 559.9 | 75.7 | 44.6 | 66.1 | 1762.3 |
| Human3R | **62.2** | **37.2** | **54.1** | **212.1** | **66.5** | **40.9** | **58.6** | **198.2** |

Table 9: **Analysis of camera-frame improvement.** Comparison with Multi-HMR on in- and out-of-distribution resolutions. Multi-HMR degrades significantly without GT intrinsics or on unseen resolutions, whereas Human3R remains stable by leveraging CUT3R's metric-scale scene priors. Notably, Human3R yields noticeable gains in local human reconstruction (pose, shape, scale, and orientation; assessed via PVE/PA-MPJPE/MPJPE) and accurate absolute human positions in the camera coordinate (MRPE).

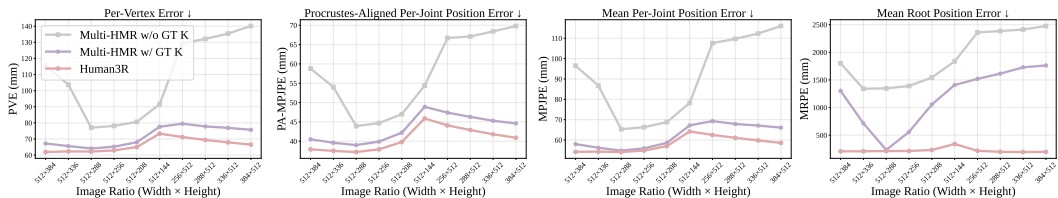

Figure 18: **Evaluation of intrinsic robustness** in human mesh reconstruction. Multi-HMR [3] performance varies when processing images at different ratios, while Human3R performs consistently well without requiring camera intrinsics, benefiting from the 3D awareness of CUT3R.

## A.6 HUMAN3R BENEFITS FROM THE 3D AWARENESS OF CUT3R

In Tab. 1, we demonstrate the advantages of Human3R in the camera coordinate space, where it shows significant improvement over Multi-HMR. To provide an in-depth analysis of how Human3R benefits from the 3D awareness of CUT3R, we disentangle the sources of improvement by examining both depth accuracy and specific spatial dimensions using the following metrics:

- **Human-centric reconstruction metrics:**
  - **Per-Vertex Error (PVE):** Root-centered error, used to assess the pose, shape, scale, and orientation correctness of the mesh vertices.
  - **Procrustes-Aligned MPJPE (PA-MPJPE):** Root-centered, scale-aligned, and rotation-aligned error, reflecting the pose and shape accuracy of SMPL joints.
  - **Mean Per-Joint Position Error (MPJPE):** Calculated using root-centered human meshes, demonstrating the pose, shape, scale, and orientation correctness of SMPL joints.
- **Scene-Aware Metric (Human absolute location in the camera coordinate):**
  - **Mean Root Position Error (MRPE)** [3]: Evaluates the camera-frame absolute pelvis location in metric scale (millimeters).

As shown in Tab. 9, Human3R benefits from CUT3R in two key aspects:

**1) Intrinsic independence via intrinsic awareness.** Leveraging the intrinsic awareness of CUT3R [97] eliminates the dependency on camera intrinsics, enabling coherent 3D human recovery from intrinsic-agnostic in-the-wild images. Specifically, Multi-HMR performs substantially worse without ground-truth (GT) intrinsics across all metrics, indicating that previous methods rely heavily on accurate camera parameters. In contrast, Human3R outperforms even the Multi-HMR baseline equipped with GT intrinsics, requiring no intrinsic inputs.

**2) Robustness to Out-of-Distribution (OOD) via metric-scale scene context.** CUT3R's metric-scale context enhances robustness against OOD image aspect ratios. While Multi-HMR (w/ GT intrinsics) suffers severe performance degradation when input aspect ratios drift out-of-distribution, as manifested by noticeable increases in human-centric metrics (i.e., PVE, PA-MPJPE, MPJPE), Human3R remains consistently strong. Furthermore, the sharp increase in the scene-aware metric (MRPE) for Multi-HMR indicates failures in placing humans in the correct spatial location under OOD conditions. Conversely, Human3R maintains accurate placement thanks to the generalizable 3D priors learned from CUT3R.

We further illustrate the behavior of both methods as data shifts from In-distribution to Out-of-distribution in Fig. 18. While Multi-HMR w/ GT intrinsics is more robust than Multi-HMR w/o GT intrinsics, its performance still fluctuates with image aspect ratios. This confirms that while

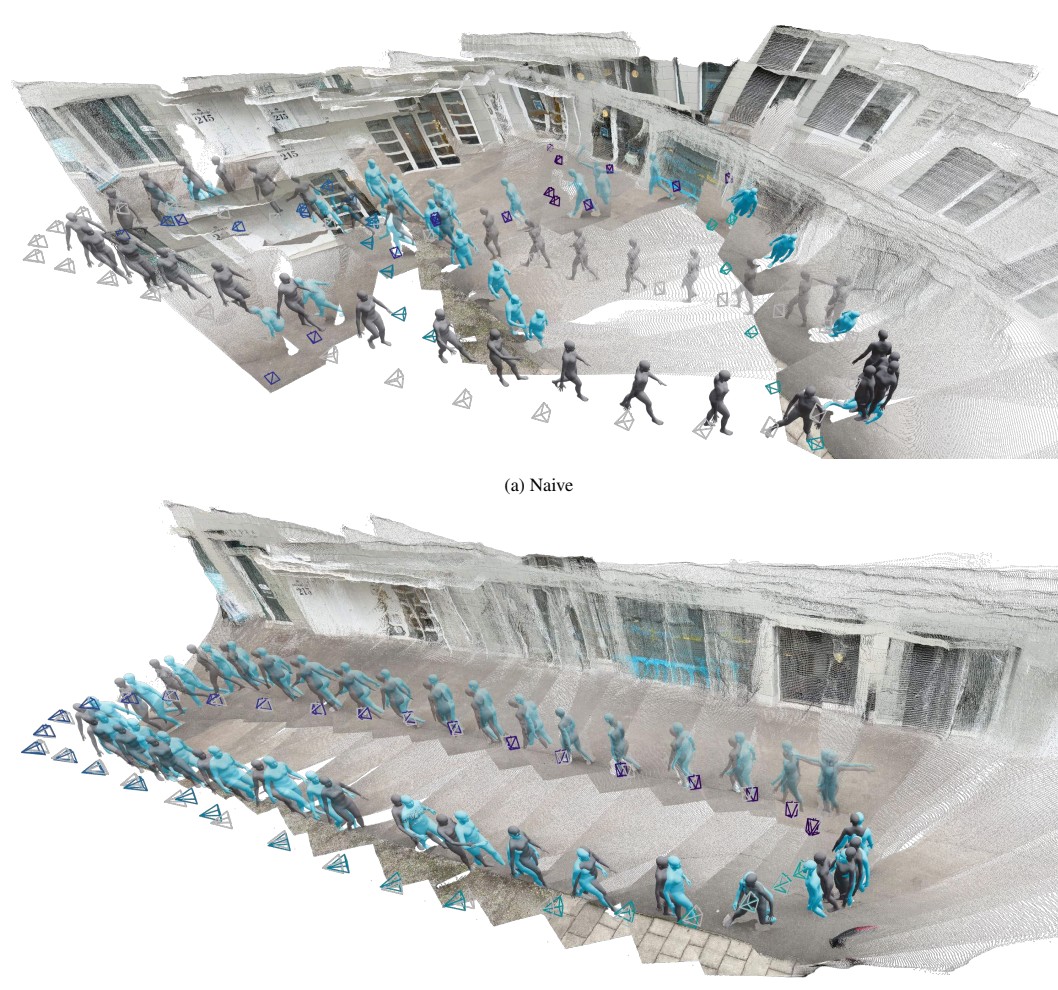

(a) Naive

(b) Ours

Figure 19: **Comparison with naive CUT3R+Multi-HMR combination** in global human motion, 3D scene reconstruction, and camera poses estimation. The colors ● and ● indicates Prediction and Ground-truth, respectively.

intrinsics aid in placing meshes [3], they are not a panacea. In contrast, Human3R exhibits consistent stability without requiring intrinsics. These findings underscore the value of using CUT3R [97] as a 4D foundation model: leveraging metric-scale scene context not only enhances intrinsic robustness but also ensures coherent recovery of 3D humans from diverse in-the-wild images.

### A.7 HUMAN3R TAKES THE BEST OF BOTH WORLDS

Human3R predicts camera poses and scenes more accurately than CUT3R (Fig. 9), reconstructs local human details better than Multi-HMR (Tab. 1, Fig. 12), and outperforms naive combinations of Multi-HMR and CUT3R on global human reconstruction (Tab. 3)—*all at once*. We visualize reconstruction results in Fig. 19 and Fig. 22. Beyond offering a unified model that jointly reasons about humans, the scene, and the camera in an online manner, Human3R runs on streaming video in real time (15 FPS), eliminating the need for separate off-the-shelf components and iterative refinement. Crucially, this efficiency does not come at the expense of accuracy. With *human prompt tuning*, our model reconstructs multiple people in a single forward pass, while implicitly reasoning about human-scene interaction (more examples on Fig. 23 and Fig. 24). Trained with only one 48G GPU for one day, it delivers substantially improved reconstructions than naive combinations of Multi-HMR and CUT3R and achieves state-of-the-art performance over task-specific baselines.

| Ablation | 3D Reconstruction | | Human Mesh Reconstruction | | | Global Human Motion | | |
|---|---|---|---|---|---|---|---|---|
| | Abs Rel ↓ | $\delta < 1.25$ ↑ | PA-MPJPE ↓ | MPJPE ↓ | PVE ↓ | WA-MPJPE ↓ | W-MPJPE ↓ | RTE ↓ |
| (a) Full model | **0.086** | **95.4** | **84.9** | **71.2** | **44.1** | **267.9** | **112.2** | **2.2** |
| (b) w/o cross attention | 0.089 | 94.5 | 93.9 | 78.9 | 47.1 | 659.8 | 256.3 | 3.7 |
| (c) w/ joint train decoder | 0.428 | 4.7 | 85.2 | 71.5 | 44.7 | 1359.3 | 621.3 | 15.3 |

Table 10: **Ablation study on multi-task performance.** Evaluations covers general 3D reconstruction (Bonn), local human mesh recovery (3DPW), and global human motion (EMDB-2). Comparing (a) our full model with (b) removing cross-attention and (c) jointly training the decoder reveals key insights: the drop in (b) verifies the crucial role of scene-derived 3D awareness via modality interdependences, and the degradation in (c) validates our *human prompt tuning*, which effectively preserves CUT3R priors while effortlessly adapting to new tasks.

## A.8 ABLATION OF CROSS-ATTENTION

To quantify the mutual benefits of joint human-scene modeling, we ablate the cross-attention mechanism between the human and scene branches to verify their interdependence.

As shown in Tab. 10 (b), the model without cross-attention performs worse across all tasks: 3D Reconstruction, Human Mesh Reconstruction, and Global Human Motion. Specifically, the degradation in Human Mesh Reconstruction aligns with our findings in Appendix A.6, confirming that Human3R largely benefits from the 3D awareness provided by the scene context. Furthermore, the decline in Global Human Motion highlights that cross-attention is essential for aligning human and scene representations within a unified metric space. Additionally, we visualize the cross-attention activation maps in Fig. 20, using the human head prompt as the query. These visualizations intuitively illustrate that the network effectively captures correlated human-scene features during reconstruction.

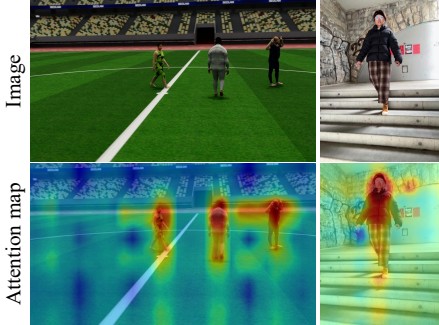

Figure 20: **Cross-attention visulization** between human head prompts $\mathbf{H}_t$ and image tokens $\mathbf{F}_t$ in the decoder.

## A.9 COMPARISON OF JOINT TRAINING STRATEGIES

To demonstrate the effectiveness of our *human prompt tuning* protocol, we conduct a joint training ablation study where we fully fine-tune the Decoder component.

As shown in Tab. 10 (c), although full fine-tuning is theoretically not restricted by a limited number of trainable parameters, it suffers from catastrophic forgetting. This is demonstrated by severe performance degradation in both 3D Reconstruction and Global Human Motion tasks. The decline occurs because CUT3R loses its prior knowledge of 3D awareness when fine-tuned on BEDLAM [5], a synthetic dataset that is relatively small-scale compared to CUT3R's large capacity. These results highlight the necessity of our efficient *human prompt tuning* protocol, which preserves pre-trained priors while effortlessly adapting to new tasks.

## A.10 ROBUSTNESS TO FEATURE RESOLUTION

Vision Transformer (ViT) encoders inherently downscale spatial resolution, which may potentially impact the granularity of human detection. To investigate this, we analyze how input resolution affects performance by resizing the encoded features with various scaling factors.

As shown in Tab. 11, we evaluate the impact of resolution by applying scaling factors ranging from ×1/8 to ×8, covering the spectrum of common image resolutions. We report performance using three metrics: **1) Precision**, which measures the percentage of detected objects that are valid human instances; **2) Recall**, which measures the percentage of actual human instances that are correctly detected; and **3) F1-score**, the harmonic mean of Precision and Recall. The results indicate that Precision remains largely stable across all scales, demonstrating that resolution changes do not induce false positives. However, Recall shows a different trend: it remains stable at higher resolutions but degrades notably

| Scaling factors | Precision ↑ | Recall ↑ | F1-score ↑ |
|---|---|---|---|
| ×1/8 | **99.4** | 55.8 | 63.8 |
| ×1/4 | 98.1 | 86.4 | 91.3 |
| ×1/2 | 98.8 | 93.7 | 96.0 |
| ×1 | 99.1 | **94.9** | **96.8** |
| ×2 | 99.0 | 94.8 | 96.7 |
| ×4 | 99.0 | 94.8 | 96.7 |
| ×8 | 99.0 | 94.8 | 96.7 |

Table 11: **Analysis on feature resolution.** We evaluate the impact of different scaling factors on detection.

at lower resolutions (e.g., $\times 1/8$). This degradation occurs because coarse feature maps can cause token collision, where multiple distinct human heads in close proximity collapse into a single feature token. This leads to missed detections (false negatives), as the model fails to distinguish individual instances within the crowded token.

Consequently, we recommend utilizing higher feature resolutions to ensure robust detection, particularly in crowded scenarios.

## B  RELATED WORKS

### B.1  GENERIC 3D RECONSTRUCTION.

3D reconstruction from RGB images has long been a fundamental challenge in computer vision. Structure-from-Motion (SfM) [1, 79, 84, 85] and SLAM [22, 26, 51, 57, 58, 114] are foundational approaches for simultaneously recovering 3D structure and camera poses. However, these methods often struggle in scenarios with small camera parallax, textureless surface, or dynamic elements, like the moving humans, and typically produce only sparse point clouds, which constrains detailed scene understanding. Moreover, their optimization-based pipelines are computationally intensive and slow, making them less suitable for real-time applications. A major breakthrough in feedforward 3D reconstruction was achieved by DUSt3R [98], which introduced an end-to-end approach that directly predicts two pixel-aligned pointmaps [7, 8, 83] from an image pair. Subsequent methods [96, 107] extended this framework to handle multiview inputs using large global attention [93], achieving state-of-the-art results in 3D point and camera pose reconstruction. However, these approaches suffer from quadratic growth in computational and memory costs, making them inherently offline: inference must be re-run over all images whenever a new frame is added. To enable online reconstruction, several works [9, 14, 95, 104, 117] introduce memory mechanisms that compress and retain information from past frames, allowing for incremental 3D reasoning. However, since these methods are not trained on dynamic datasets [113] or do not explicitly disentangle static scenes from dynamic humans [11], their performance degrades when processing videos with moving people. A promising advance is the recurrent deep 4D reconstruction foundation model, CUT3R [97], which is trained on both static and dynamic datasets. CUT3R achieves feed-forward 4D reconstruction by maintaining a persistent internal state that encodes the spatiotemporal history of both scenes and humans, incrementally updating this state as new observations arrive. This recurrent formulation enables efficient processing of long sequences with linear computational complexity, while keeping inference memory usage consistently low. Building on this success, we leverage the spatiotemporal priors learned by CUT3R to enable online holistic 4D reconstruction, reasoning jointly not only the 3D scene and camera poses, but also the multi-person human body mesh sequences (parameterized with SMPL-X [64]), in the world frame, at a real-time inference speed.

## C  TRAINING DETAILS

We freeze all weights of pretrained CUT3R and Multi-HMR encoder, and fine-tune the human-related modules (i.e., $\text{Head}_{\text{projection}}$, $\text{Head}_{\text{human}}$, $\text{MLP}_{\text{head}}$ and $\text{MLP}_{\text{mask}}$) on BEDLAM [5]. This dataset provides 3D scene depth and SMPL-X meshes, with 1–10 people per scene, captured from diverse known camera viewpoints. Following CUT3R, we exclude BEDLAM sequences where the environment is represented by a panoramic HDRI image, resulting in 5,000 sequences for training and 1,000 for validation, with each sequence averaging 30 frames. For each iteration, we randomly sample 4 frames from each sequence and train Human3R with a batch size of 8, using variable aspect ratios and resizing images so that the longer side is 512 pixels. All MLP networks are implemented as 2 linear layers with GELU activation [34]. Each *human prompt*, a 768-dimensional vector, is concatenated with the camera and image tokens along the token dimension. We use the AdamW optimizer [55] with an initial learning rate of $1 \times 10^{-4}$, employing linear warmup followed by cosine decay. We train our model on a single NVIDIA 48GB GPU within one day.

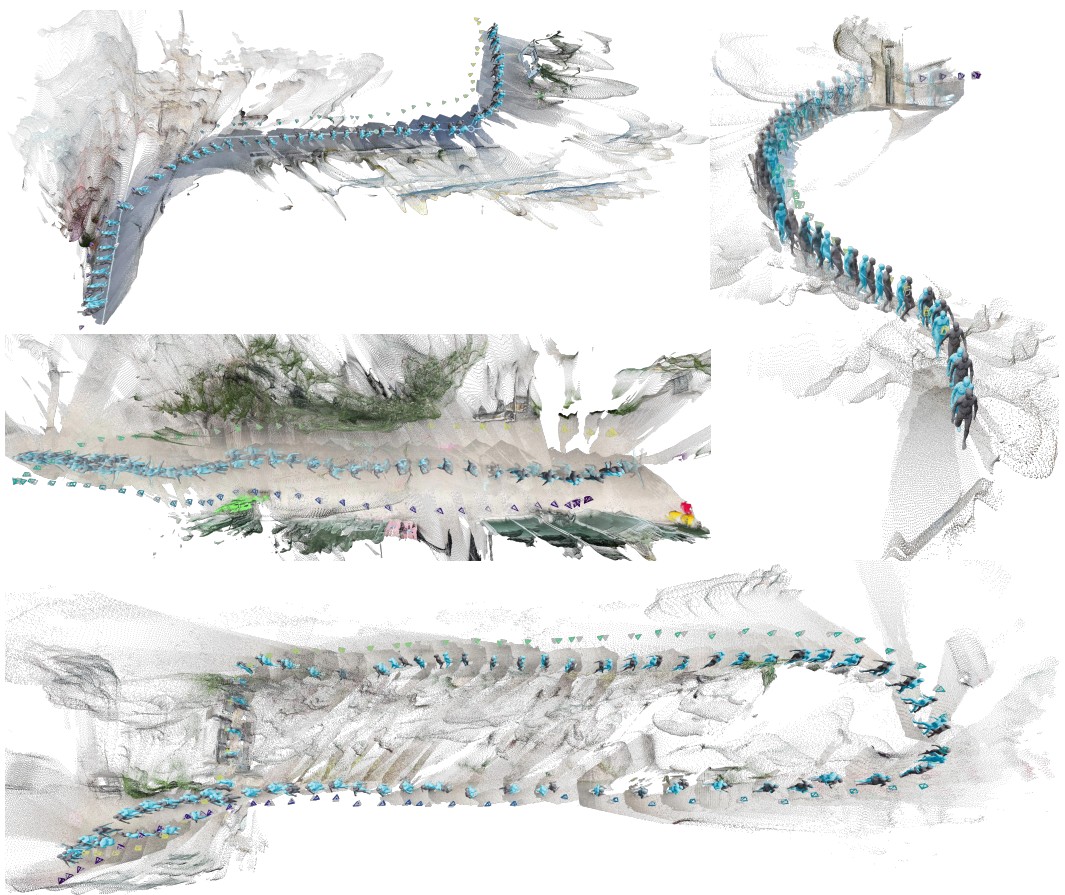

Figure 22: **Qualitative comparison** in global human motion, 3D scene reconstruction, and camera poses estimation of ● our prediction against ● ground-truth on EMDB dataset (subset 1) [39].

## D  FAILURE CASES & FUTURE WORK

Human3R implicitly models human interactions (Fig. 21, left) but does not yet resolve them (Fig. 21, middle), and it has not matched strong offline methods (e.g., JOSH [53]) in reconstruction accuracy. Iterative optimization—though slower and more memory-intensive—better constrains interpenetration, physics, and contacts. Human3R can therefore serve as an effective initialization for applications that demand high accuracy. While Human3R shows a clear boost in real-time human-scene reconstruction, its design space remains largely unexplored. Fig. 21 (right) highlights a vast opportunity to develop more expressive architectures for handling human-object interactions and

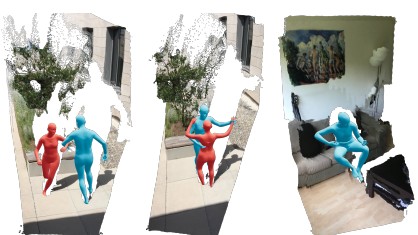

(a) Example    (b) Interaction   (c) Dynamic object
Figure 21: **Failure cases.** (a) Successful human-human interaction; (b) Interaction failures with human-human interpenetration; (c) Inability to model dynamic objects.

moving toward *everything*. We hope that this work will motivate future research to revisit the task of dynamic human, animal, and object from a real-time, online, end-to-end perspective.

## E  USE OF LARGE LANGUAGE MODELS

We used a large language model to assist with copy editing—grammar checking, wording suggestions, and minor style and clarity improvements—after the scientific content, methodology, analyses, and conclusions had been written by the authors.

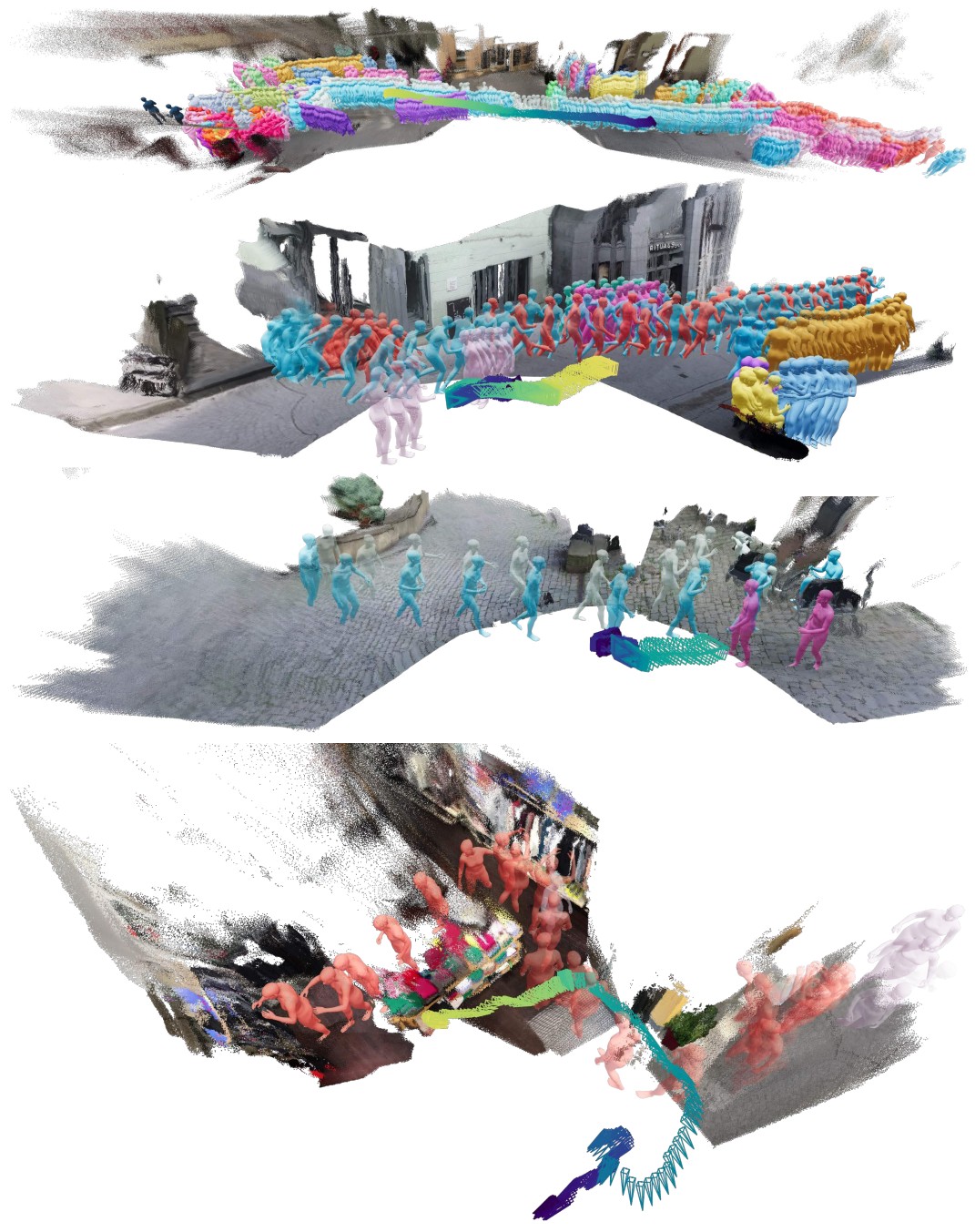

Figure 23: **Qualitative 4D human-scene reconstruction results** on the 3DPW dataset [94]. Given video captured from a moving camera, Human3R performs online reasoning about global human motion, the surrounding environment, and camera poses *all at once*.

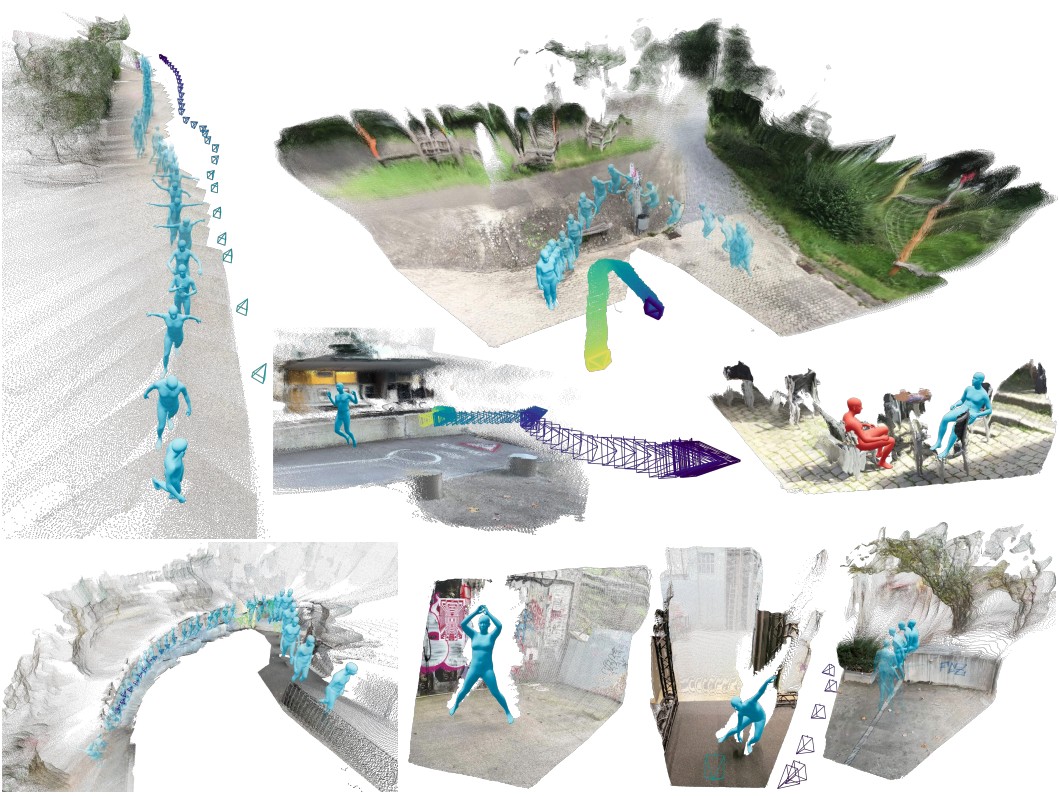

Figure 24: **Qualitative 4D human-scene reconstruction results** on the EMDB dataset [39]. Given video captured from a single camera, Human3R performs online reasoning about global human motion, the surrounding environment, and camera poses *all at once*.

