# OpenReview forum: "Human3R: Everyone Everywhere All at Once"
_ICLR.cc/2026/Conference — ICLR 2026 Poster_

### Official Review · Reviewer_N5HW · 2025-10-26

**Soundness:** 2
**Presentation:** 2
**Contribution:** 3
**Rating:** 4
**Confidence:** 5

**Summary:**

This paper addresses the task of reconstructing 3D human motion, camera motion, and scene from monocular videos. Unlike previous multi-stage approaches, it proposes a unified single-shot framework that jointly reasons about humans, cameras, and the surrounding scene. The framework is built upon the 4D reconstruction model CUT3R, which recurrently estimates metric-scale depth and camera poses in an online fashion. To enhance 3D human motion understanding, the method integrates human pose tokens extracted by Multi-HMR with image features from CUT3R’s backbone. During training, the CUT3R weights are frozen, while only the layers responsible for fusing the Multi-HMR tokens and regressing SMPL-X parameters are fine-tuned on the BEDLAM dataset. The proposed model is evaluated on 3DPW and EMDB (subset1) in camera coordinates, and on EMDB (subset2) and RICH in global coordinates. Experimental results show that the model achieves performance comparable to existing HMR methods in camera coordinates and, leveraging CUT3R’s robust global reasoning, delivers promising results in global coordinates.

**Strengths:**

1. The paper proposes to addresses the challenge of fusing CUT3R features with Multi-HMR features during both training and inference by introducing pixel-aligned feature extraction and Test-Time Sequence Length Adaptation. This enables a unified multi-task learning framework under limited training data, advancing the integration from feature-level fusion beyond the previous result-level fusion used in methods like GVHMR.

2. The experiments reveal several valuable insights. As shown in Table 1, directly integrating CUT3R’s 4D reconstruction features with the human motion features extracted by Multi-HMR via pixel-aligned extraction improves HMR performance in the camera coordinate system. The ablation studies in Section 4.4 further analyze the contribution of each feature type, which is helpful for understanding their respective effects.

3. The visualization results presented are promising and engaging, likely to attract readers’ interest and confidence in the method’s potential.

**Weaknesses:**

1. The proposed framework estimates global camera poses using CUT3R and fuses them with human features extracted by Multi-HMR to recover human motion. However, the overall computational complexity of the open-source implementation is quite similar to that of GVHMR, which also integrates visual odometry and 2D pose estimation (using different backbones corresponding to the variants described in this paper). Both methods share similar design philosophies and architectural concepts. As shown in Table 2, their performance metrics are also close, while Table 1 reveals a noticeable performance gap. One possible reason for this discrepancy could be the limited number of trainable layers — the proposed approach does not jointly train the Decoder component illustrated in Figure 4, which may restrict performance. Therefore, GVHMR serves as a strong and conceptually comparable competitor, and such a comparison could better reveal the advantages and limitations of the paper’s “low-training” design philosophy.

2. The presentation of the Human Prior module in Figure 4 may cause confusion among readers. The human-prior token used in this work is derived from the pixel-aligned feature vectors between CUT3R’s image features and the image features extracted by Multi-HMR’s backbone. However, Figure 4 omits a clear description of how these tokens are obtained, which may lead readers to mistakenly assume that a single backbone is used for feature extraction. However, both CUT3R and Multi-HMR backbones independently extract their own image features before being fused at corresponding pixel positions.

3. Some of the claimed contributions lack sufficient empirical evidence, particularly the statement that the proposed method “operates on streaming video at real-time speed (15 FPS) without compromising accuracy.” I tested the released implementation on an RTX 4090 GPU and found that the inference speed — when extracting CUT3R and Multi-HMR features online — falls far short of real-time, even significantly below the 5 FPS reported in Table 4 for the largest backbone. It remains unclear how the ViT-B version used in the real-time mode performs and whether accuracy degradation occurs. In addition, the latency details should be explicitly discussed.

**Questions:**

1. It would be better to provide a more in-depth analysis of Table 1, clarifying whether the observed improvement in the camera coordinate system (compared to Multi-HMR alone) primarily comes from depth accuracy or from other specific spatial dimensions.
2. It would be more appropriate to revise the contribution statements, especially regarding real-time performance claims, to report and justify the corresponding runtime metrics and latency with clear condition.

---

> ### Author Response · Authors · 2025-11-30
> **Official Response by Authors (1/4)**
>
> We appreciate the feedback. We revise the manuscript and address each point below:
>
> > **Q1:  (Part 1\) The proposed framework estimates global camera poses using CUT3R and fuses them with human features extracted by Multi-HMR to recover human motion. However, the overall computational complexity of the open-source implementation is quite similar to that of GVHMR, which also integrates visual odometry and 2D pose estimation (using different backbones corresponding to the variants described in this paper). Both methods share similar design philosophies and architectural concepts.**
>
> ### **Part 1: Conceptual differences & Computational complexity**
>
> We would like to clarify that Human3R can directly be used for **online inference** and **Multiple Humans \+ 3D Scene**, whereas GVHMR and TRAM can only estimate single human and multiple human respectively, and no scene. In addition, Human3R does not require external modules (Bounding box detection, Tracking , 2D pose estimation, SLAM, Metric depth estimation) and is therefore an **end-to-end and scalable** method.
>
> **Conceptual Comparison:**
>
> |  | GVHMR | TRAM | Human3R |
> | :---- | :---: | :---: | :---: |
> | 3D scene | ✗ | ✗ | **✓** |
> | Multi-person | ✗ | **✓** | **✓** |
> | End-to-end learning | ✗ | ✗ | **✓** |
> | Online | ✗ | ✗ | **✓** |
> | FPS | \~5 | \~1 | **5\~15** |
>
> Furthermore, we thoroughly compare the time of each component for those methods in the revised manuscript (`Tab.5`, `Tab.6`, `Tab.7`, and `L.1173-L.1214`).
>
> **Summary of results:** Human3R achieves **5\~15 FPS** (with ViT-S/B/L), outperforming existing methods:
>
> * **vs. Human-only methods:** Unlike single-person method GVHMR (**4.88 FPS**) and multi-person method TRAM (**0.86 FPS**), Human3R recovers the full 3D scene and camera trajectory, not just humans.
> * **vs. Joint reconstruction:** Regarding JOSH, which jointly reconstructs humans and scenes, we exclude a detailed breakdown as the code is not yet public. Their reported overall runtime on an NVIDIA RTX 4090 GPU is **\~0.8 FPS**.
>
> **We report the model inference time** **on an NVIDIA RTX 4090 GPU for all methods.** Next, we expand runtime breakdown comparison:
>
> **1\. Offline baselines (GVHMR & TRAM) (Added `Tab.5`, `Tab.6`)**
>
> * **GVHMR (Single-human):** GVHMR requires four separate preprocessing steps to extract four feature types: bounding box detection, 2D keypoints, image features, and camera poses. Notably, these costs only apply to single-person reconstruction, as it does not support multi-person scenarios. The complete offline process operates at 4.88 FPS.
>
>     | Methods | GVHMR (Single-human) |
>     | :---- | :---: |
>     | YOLOv8 (Detection) | 0.041 |
>     | ViTPose (2D Keypoint) | 0.067 |
>     | Vit Feature Extraction | 0.059 |
>     | DPVO (SLAM) | 0.037 |
>     | HMR | 0.001 |
>     | **Total** | 0.205s (**4.88 FPS**) |
>
> * **TRAM (Multi-human):** TRAM requires computationally intensive preprocessing, including detection and tracking, segmentation, camera pose estimation, and metric depth estimation. Although it supports multi-person reconstruction, its pipeline is limited to offline operation at 0.86 FPS.
>
>     | Methods | TRAM (Multi-human) |
>     | :---- | :---: |
>     | DEVA (Detection\&Track) | 0.3617 |
>     | SAM (Segmentation) | 0.1757 |
>     | DROID-SLAM | 0.0463 |
>     | ZoeDepth (Metric Depth) | 0.0120 |
>     | Global Alignment | 0.0049 |
>     | VIMO (HMR) | 0.5581 |
>     | **Total** | 1.1587s (**0.86 FPS**) |
>
> **Challenges:** Dependencies associated with off-the-shelf estimators often impose synchronization barriers, which induce cumulative errors and hinder online inference. Furthermore, they face scalability bottlenecks.
>
> **2\. Efficiency of Human3R (Ours) (Added `Tab.7`)**
> To eliminate these dependencies, we design a unified framework **for online inference**, leveraging generalizable priors within **an end-to-end, data-driven model**.
>
> * **Streamlined Architecture:** Guided by the design principles of simplicity and scalability, Human3R adopts a streamlined architecture composed exclusively of an Encoder, a Decoder, and Heads.
> * **Computation Analysis**: The cost distribution among the Encoder, Decoder, and Heads is approximately 1:2:2. The latency of the Multi-HMR encoder varies depending on model size (ranging from ViT-S/672 to ViT-L/896). As shown in `Appendix A.5`, the ViT-S/672 variant offers an optimal accuracy-speed trade-off, achieving 15.87 FPS.
>
>     | Modules | Human3R (Multi-human & Scene) |  |  |  |
>     | :---- | :---: | :---: | :---: | :---: |
>     |  | ViT-S/672 | ViT-B/672 | ViT-L/672 | ViT-L/896 |
>     | Encoder | 0.012 | 0.012 | 0.012 | 0.012 |
>     | Multi-HMR Encoder | 0.011 | 0.023 | 0.057 | 0.134 |
>     | Decoder | 0.019 | 0.019 | 0.019 | 0.019 |
>     | Heads | 0.021 | 0.021 | 0.021 | 0.021 |
>     | **Total (s)** | **0.063** | 0.075 | 0.109 | 0.186 |
>     | **FPS** | **15.87** | 13.33 | 9.17 | 5.38 |

---

> ### Author Response · Authors · 2025-11-30
> **Official Response by Authors (2/4)**
>
> > **Q1 (Part 2\) As shown in Table 2, their performance metrics are also close, while Table 1 reveals a noticeable performance gap. One possible reason for this discrepancy could be the limited number of trainable layers — the proposed approach does not jointly train the Decoder component illustrated in Figure 4, which may restrict performance. Therefore, GVHMR serves as a strong and conceptually comparable competitor, and such a comparison could better reveal the advantages and limitations of the paper’s “low-training” design philosophy.**
>
> ### **Part 2: Ablation on Joint Training Decoder**
>
> Following your suggestion, **we conduct an ablation fully fine-tuning the Decoder component** in the revised manuscript (**Added** **`Tab.10c`** and `L.1378–L.1389`).
>
> **Catastrophic Forgetting:** As shown in table below, although joint training has theoretically more trainable parameters, it leads to significant performance degradation in both Global Human Motion and 3D Reconstruction tasks.
>
> **Analysis:** The decline occurs because  Decoder (CUT3R) loses its prior knowledge of 3D awareness when fine-tuned on the relatively smaller-scale BEDLAM dataset (compared to CUT3R's large network capacity). This validates our *"human prompt tuning"* philosophyn—freezing the pre-trained scene priors while effortlessly adapting to new tasks.
>
> | Ablation | Scene |  | Local Human |  |  | Global Human |  |  |
> | :---- | :---: | :---: | :---: | :---: | :---: | :---: | :---: | :---: |
> |  | Abs Rel $\\downarrow$ | $\\delta \< 1.25 \\uparrow$ | PA-MPJPE $\\downarrow$ | MPJPE $\\downarrow$ | PVE $\\downarrow$ | WA-MPJPE $\\downarrow$ | W-MPJPE $\\downarrow$ | RTE $\\downarrow$ |
> | Full model | **0.086** | **95.4** | **84.9** | **71.2** | **44.1** | **267.9** | **112.2** | **2.2** |
> | w/ joint train decoder | 0.428 | 4.7 | 85.2 | 71.5 | 44.7 | 1359.3 | 621.3 | 15.3 |
>
> ---
>
> > **Q2: The presentation of the Human Prior module in Figure 4 may cause confusion among readers. The human-prior token used in this work is derived from the pixel-aligned feature vectors between CUT3R’s image features and the image features extracted by Multi-HMR’s backbone. However, Figure 4 omits a clear description of how these tokens are obtained, which may lead readers to mistakenly assume that a single backbone is used for feature extraction. However, both CUT3R and Multi-HMR backbones independently extract their own image features before being fused at corresponding pixel positions.**
>
> **We resolve the Figure 4 ambiguity by explicitly illustrating the dual-encoder design (CUT3R and Multi-HMR)** in the revised manuscript (`Fig.6`, `L.223`). Main updates are as follows:
>
> **1\. New Visualization (Added `Fig.6`):** To address the concern about how tokens are obtained, we have added **Figure 6** (*"Integration of human priors via the human-centric Multi-HMR ViT-DINO image encoder"*) with a clear workflow description:
>
> * **Separate encoders** used by CUT3R and Multi-HMR.
> * **Explicit flow** of concatenating pixel-aligned features before projection by the MLP.
>
> **2\. Revised Figure 4 Caption (`L.223`)** To clarify the token extraction process directly within the figure's context, we have revised the caption of Figure 4 (Line 223\) to explicitly state the use of the separate Multi-HMR encoder.
>
> **Original:** "Each detected head token, concatenated with a human prior token from Multi-HMR ViT-DINO feature, is projected into a human prompt."
>
> **Revised:** "Each detected head token is concatenated with a human prior token—**sampled from a separate Multi-HMR ViT-DINO encoder at the corresponding pixel coordinates**—and subsequently projected into a human prompt."

---

> ### Author Response · Authors · 2025-11-30
> **Official Response by Authors (3/4)**
>
> > **Q3: Some of the claimed contributions lack sufficient empirical evidence, particularly the statement that the proposed method “operates on streaming video at real-time speed (15 FPS) without compromising accuracy.” I tested the released implementation on an RTX 4090 GPU and found that the inference speed — when extracting CUT3R and Multi-HMR features online — falls far short of real-time, even significantly below the 5 FPS reported in Table 4 for the largest backbone. It remains unclear how the ViT-B version used in the real-time mode performs and whether accuracy degradation occurs. In addition, the latency details should be explicitly discussed.**
>
> Thank you for your suggestion. We believe there is an **important misunderstanding** regarding the scope of the latency measurement. **The 15 FPS claim refers to the model inference time (forward pass)**, whereas the `evaluation.py` includes significant time cost from ground-truth data processing, computing evaluation metrics, visualization, and I/O operations.
>
> To clear up this confusion, **we provide a comprehensive runtime analysis** in the revised manuscript (added `Tab.4` and `Tab.8` in `Appendix A.4`, `Appendix A.5`).
>
> **1\. Clarification: Inference pipeline** **vs. Evaluation pipeline** The reviewer’s observed latency includes auxiliary steps (loading, saving, rendering) that are not part of the deployment runtime.
>
> * **Action:** We will release a streamlined `inference_only.py` script to allow reproducibility of the raw model speed.
> * **Benchmark Setup:** All reported speeds are measured on an NVIDIA RTX 4090 GPU with dual Intel Xeon Gold 6530 CPUs (`L.1150`).
>
> **2\. Comprehensive Runtime Analysis (Added `Tab.4`):** We have added Table 4 to benchmark runtime across diverse datasets and backbones. As shown below, the **ViT-S/672** variant indeed supports real-time applications (\~15 FPS), while larger models trade speed for detail:
>
> | Models | FPS |  |  |  |  |  |
> | :---: | :---: | :---: | :---: | :---: | :---: | :---: |
> |  | 3DPW (288$\\times$512) | BEDLAM (512$\\times$288) | RICH (512$\\times$368) | EMDB (384$\\times$512) | Bonn (512$\\times$384) | TUM-D (512$\\times$384) |
> | Human3R w/ ViT-S/672 | **15.87** | 15.64 | 14.28 | 13.75 | 13.65 | 13.59 |
> | Human3R w/ ViT-B/672 | **13.33** | 12.69 | 11.89 | 11.68 | 11.67 | 12.41 |
> | Human3R w/ ViT-L/672 | **9.17** | 8.73 | 8.38 | 8.27 | 8.27 | 8.61 |
> | Human3R w/ ViT-L/896 | **5.38** | 5.3 | 5.15 | 5.09 | 5.06 | 5.15 |
>
> **3\. Accuracy vs. Speed Trade-off (`Tab.8`)** Addressing the query on ViT-B performance and potential degradation, we added Table 8 to explicitly analyze the trade-off.
>
> * **Real-Time Tier:** **ViT-S** (15 FPS) offers a strong balance for global motion estimation (WA-MPJPE 129.9, RTE 2.2).
> * **High-Fidelity Tier:** **ViT-L** (5-7 FPS) provides more detailed human-mesh reconstruction (WA-MPJPE 112.2, RTE 2.2), suitable for application requiring fine-grained pose and shape.
>
> | Ablations | Local Human |  |  | Global Human |  |  | Runtime |
> | :---: | :---: | :---: | :---: | :---: | :---: | :---: | :---: |
> |  | PA-MPJPE$\\downarrow$ | MPJPE$\\downarrow$ | PVE$\\downarrow$ | WA-MPJPE$\\downarrow$ | W-MPJPE$\\downarrow$ | RTE$\\downarrow$ | FPS$\\uparrow$ |
> | Human3R w/ ViT-S/672 | 56.1 | 87.8 | 103.1 | 129.9 | 314.2 | 2.2 | **15** |
> | Human3R w/ ViT-B/672 | 49.3 | 79.6 | 94.3 | 122.1 | 292.9 | 2.2 | 11 |
> | Human3R w/ ViT-L/672 | 48.5 | 83.1 | 96.7 | 113.6 | 291.7 | 2.2 | 7 |
> | Human3R w/ ViT-L/896 | **44.1** | **71.2** | **84.9** | **112.2** | **267.9** | **2.2** | 5 |
>
> **Conclusion***:* The proposed method covers a spectrum from real-time efficiency (ViT-S) to high-fidelity reconstruction (ViT-L), verifying our claims.

---

> ### Author Response · Authors · 2025-11-30
> **Official Response by Authors (4/4)**
>
> > **Q4: It would be better to provide a more in-depth analysis of Table 1, clarifying whether the observed improvement in the camera coordinate system (compared to Multi-HMR alone) primarily comes from depth accuracy or from other specific spatial dimensions.**
>
> We thank the reviewer for this constructive suggestion. **To disentangle the sources of improvement in the camera coordinates** (benefit from depth accuracy or other spatial dimensions), **we provide an in-depth analysis** in the revised manuscript (`Tab.9`,  `Fig.18` in `Appendix A.6`).
>
> We examined improvement across two dimensions: **Human-centric metrics** (PVE, PA-MPJPE, MPJPE for pose/shape) and **Scene-aware metrics** (MRPE for human absolute depth/location).
>
> **1\. Detailed Breakdown of Improvements (Added `Tab.9`)** As shown below, Human3R outperforms Multi-HMR across **all dimensions**, significantly enhancing both local pose fidelity and global spatial placement.
>
> | Methods | In-distribution |  |  |  | Out-of-distribution |  |  |  |
> | ----- | :---: | :---: | :---: | :---: | :---: | :---: | :---: | :---: |
> |  | PVE $\\downarrow$ | PA-MPJPE $\\downarrow$ | MPJPE $\\downarrow$ | MRPE $\\downarrow$ | PVE $\\downarrow$ | PA-MPJPE $\\downarrow$ | MPJPE $\\downarrow$ | MRPE $\\downarrow$ |
> | Multi-HMR w/o GT intrinsic | 77 | 43.9 | 65.3 | 1347.5 | 140.1 | 69.8 | 116 | 2475.7 |
> | Multi-HMR w/ GT intrinsic | 64.1 | 39 | 54.8 | 559.9 | 75.7 | 44.6 | 66.1 | 1762.3 |
> | Human3R | **62.2** | **37.2** | **54.1** | **212.1** | **66.5** | **40.9** | **58.6** | **198.2** |
>
> **2\. Our Key Advantages:** The results highlight three advantages inherent to Human3R's design:
>
> **Advantage 1: Significant gain in depth accuracy (Scene-aware)**
> Responding directly to the reviewer's request, a major portion of the improvement indeed comes from depth reasoning. The **MRPE (Mean Root Position Error)**, which measures absolute location in the camera frame, is reduced drastically from 559.9mm (Multi-HMR w/ GT intrinsic) to **212.1mm** (Human3R). This confirms that Human3R effectively leverages metric-scale scene context to resolve absolute depth ambiguities.
>
> **Advantage 2: Improved pose and shape fidelity (Human-centric)**
> Beyond depth, the improvement also extends to the local human mesh. Even compared to Multi-HMR with ground-truth intrinsics input, Human3R achieves lower **PVE** (62.2 vs 64.1) and **PA-MPJPE** (37.2 vs 39.0). This demonstrates that our method's scene understanding capabilities help resolve scale and orientation ambiguities, directly refining the intrinsic human shape and pose estimation.
>
> **Advantage 3: Intrinsic independence & OOD robustness**
> Crucially, Human3R achieves superior robustness compared to baselines:
>
> * **Intrinsic-Agnostic:** Previous methods rely heavily on camera intrinsics (performance drops significantly without them: MRPE 559.9 $\\rightarrow$ 1347.5). Human3R achieves state-of-the-art results **without requiring any intrinsic inputs**.
> * **OOD Robustness:** As visualized in **`Fig.18`**, when image aspect ratios drift Out-of-Distribution, Multi-HMR suffers severe degradation (PVE spikes to 75.7 even with GT intrinsic). In contrast, Human3R remains stable (PVE 66.5), proving that our framework successfully learns a robust prior resilient to varied in-the-wild settings.
>
> ---
>
> > **Q5: It would be more appropriate to revise the contribution statements, especially regarding real-time performance claims, to report and justify the corresponding runtime metrics and latency with clear condition.**
>
> **We have revised the contribution statement in the updated manuscript (`L.071`)** to explicitly clarify the hardware conditions and the trade-off between speed and accuracy:
>
> **Original:** “In contrast to prior work with iterative refinement, our method runs in an online fashion, operates on streaming video at real-time speed (15 FPS) without compromising accuracy..”
>
> **Revised:** “In contrast to prior work with iterative refinement, our method runs in an online fashion. *Specifically, our lightweight model operates on streaming video at real-time speeds (15 FPS on an RTX 4090\) while maintaining competitive performance. It also offers configurable scalability, allowing for higher-fidelity reconstruction with larger backbones.*”
>
> Furthermore, to **empirically justify these claims**, we have added extensive evaluations:
>
> 1. **Runtime Benchmark:** FPS comparisons of diverse backbones across six datasets and resolutions (`Tab.4`, `L.1150-L.1172`).
> 2. **Latency Breakdown:** A detailed analysis of component-wise latency (`Tab.7`, `L.1199-L.1214`).
> 3. **Accuracy-Speed Trade-off:** An analysis of reconstruction accuracy versus inference speed (`Tab.8`, `L.1227-L.1241` in `Appendix A.5`).

---

### Official Review · Reviewer_HBKC · 2025-10-30

**Soundness:** 3
**Presentation:** 3
**Contribution:** 4
**Rating:** 8
**Confidence:** 3

**Summary:**

This paper introduces Human3R, a unified and feed-forward framework for online 4D human–scene reconstruction from monocular videos. In contrast to traditional multi-stage approaches, Human3R performs joint reconstruction of multiple SMPL-X bodies, dense 3D scenes, and camera trajectories within a single forward pass. Built upon pretrained CUT3R and enhanced through fine-tuning, the model achieves real-time performance while maintaining strong accuracy and efficiency across diverse reconstruction tasks.

**Strengths:**

1. Soundness: The paper presents a simple yet powerful idea, effectively leveraging the rich priors of CUT3R to achieve comprehensive human–scene reconstruction.

2. Ablation studies: The ablation studies are insightful and thorough, particularly those in Table 3, which clearly demonstrate how refining the pretrained priors leads to consistently improved results. The experiments are well-designed and highlight the robustness of the proposed framework.

**Weaknesses:**

1. Computation Comparison: While the model claims real-time, all-in-one performance, a computation and runtime comparison with other methods would strengthen the argument for its efficiency.

2. Cross Attention: Although the method jointly models human and scene components, it remains unclear how much each modality benefits the other. An ablation removing the cross-attention between humans and scenes could verify this interdependence. Furthermore, visualizing cross-attention activation maps could help confirm whether the network indeed captures correlated human–scene features during reconstruction.

**Questions:**

1. The ViT encoder from CUT3R may downscale spatial resolution, which might influence human detection and segmentation accuracy. Have the authors analyzed how input or feature resolution affects performance? It would be valuable to evaluate this by resizing the encoded features with various (i, j) scaling factors.

2. In scenarios where multiple humans are heavily occluded and occupy the same head token, how does the model differentiate and handle such cases? A clarification or visualization would be helpful to understand its robustness under severe occlusion.

---

> ### Author Response · Authors · 2025-11-30
> **Official Response by Authors (1/3)**
>
> We appreciate the feedback. We revise the manuscript and address each point below:
>
> > **Q1: Computation Comparison: While the model claims real-time, all-in-one performance, a computation and runtime comparison with other methods would strengthen the argument for its efficiency.**
>
> We thank the reviewer for this constructive suggestion. We follow your suggestion to **add a detailed runtime comparison against SOTA global HMR methods (GVHMR, TRAM)** in the revised manuscript (`Tab.5`, `Tab.6`, `Tab.7`, and `L.1173-L.1214`).
>
> **Summary of results:** Human3R achieves **5\~15 FPS** (with ViT-S/B/L), outperforming existing methods:
>
> * **vs. Human-only methods:** Unlike single-person method GVHMR (**4.88 FPS**) and multi-person method TRAM (**0.86 FPS**), Human3R recovers the full 3D scene and camera trajectory, not just humans.
> * **vs. Joint reconstruction:** Regarding JOSH, which jointly reconstructs humans and scenes, we exclude a detailed breakdown as the code is not yet public. Their reported overall runtime on an NVIDIA RTX 4090 GPU is **\~0.8 FPS**.
>
> **We report the model inference time** (average runtime per frame in seconds) **on an NVIDIA RTX 4090 GPU for all methods.** Next, we expand runtime breakdown comparison in detail:
>
> **1\. Offline baselines (GVHMR & TRAM) (Added `Tab.5`, `Tab.6`)**
>
> * **GVHMR (Single-human):** As shown in the table below, GVHMR requires four separate preprocessing steps to extract four feature types: bounding box detection (YOLOv8), 2D keypoints (ViTPose), image features (ViT), and camera poses (DPVO); then MLPs map these features to a common dimension, followed by element-wise summation to generate per-frame tokens for the HMR network. Notably, these costs only apply to single-person reconstruction, as it does not support multi-person scenarios. The complete offline process operates at 4.88 FPS.
>
>     | Methods | GVHMR (Single-human) |
>     | :---- | :---: |
>     | YOLOv8 (Detection) | 0.041 |
>     | ViTPose (2D Keypoint) | 0.067 |
>     | Vit Feature Extraction | 0.059 |
>     | DPVO (SLAM) | 0.037 |
>     | HMR | 0.001 |
>     | **Total** | 0.205s **(4.88 FPS)** |
>
> * **TRAM (Multi-human):** TRAM requires computationally intensive preprocessing, including detection and tracking (DEVA), segmentation (SAM), camera pose estimation (DROID-SLAM), and metric depth estimation (ZoeDepth); then these intermediate results are integrated via global alignment, followed by the HMR network for local human reconstruction. Finally, all camera-space estimations are transformed into the world coordinate using globally aligned camera poses. As detailed below, although it supports multi-person reconstruction, its pipeline is limited to offline operation at 0.86 FPS.
>
>     | Methods | TRAM (Multi-human) |
>     | :---- | :---: |
>     | DEVA (Detection\&Track) | 0.3617 |
>     | SAM (Segmentation) | 0.1757 |
>     | DROID-SLAM | 0.0463 |
>     | ZoeDepth (Metric Depth) | 0.0120 |
>     | Global Alignment | 0.0049 |
>     | VIMO (HMR) | 0.5581 |
>     | **Total** | 1.1587s (**0.86 FPS**) |
>
> **Challenges:** Dependencies associated with off-the-shelf estimators often impose synchronization barriers, which induce cumulative errors and hinder online inference. Furthermore, they face scalability bottlenecks like the *tabula rasa* problem.
>
> **2\. Efficiency of Human3R (Ours) (Added `Tab.7`)**
> To eliminate these dependencies, we design a unified framework **for online inference**, leveraging generalizable priors within **an end-to-end, data-driven model**.
>
> * **Streamlined Architecture:** Guided by the design principles of simplicity and scalability, Human3R adopts a streamlined architecture composed exclusively of an Encoder, a Decoder, and Heads. We also incorporate a Multi-HMR ViT image encoder (without the HPH decoder), initialized with pre-trained DINO weights and fully fine-tuned on human-specific datasets, to serve as a robust human prior.
> * **Computation Analysis**: As detailed in table below, the cost distribution among the Encoder, Decoder, and Heads is approximately 1:2:2. The latency of the Multi-HMR encoder varies depending on model size (ranging from ViT-S/672 to ViT-L/896). As shown in `Appendix A.5`, the ViT-S/672 variant offers an optimal accuracy-speed trade-off, achieving 15.87 FPS.
>
>     | Modules | Human3R (Multi-human & Scene) |  |  |  |
>     | :---- | :---: | :---: | :---: | :---: |
>     |  | ViT-S/672 | ViT-B/672 | ViT-L/672 | ViT-L/896 |
>     | Encoder | 0.012 | 0.012 | 0.012 | 0.012 |
>     | Multi-HMR Encoder | 0.011 | 0.023 | 0.057 | 0.134 |
>     | Decoder | 0.019 | 0.019 | 0.019 | 0.019 |
>     | Heads | 0.021 | 0.021 | 0.021 | 0.021 |
>     | **Total (s)** | **0.063** | 0.075 | 0.109 | 0.186 |
>     | **FPS** | **15.87** | 13.33 | 9.17 | 5.38 |

---

> ### Author Response · Authors · 2025-11-30
> **Official Response by Authors (2/3)**
>
> > **Q2: Cross Attention: Although the method jointly models human and scene components, it remains unclear how much each modality benefits the other. An ablation removing the cross-attention between humans and scenes could verify this interdependence. Furthermore, visualizing cross-attention activation maps could help confirm whether the network indeed captures correlated human–scene features during reconstruction.**
>
> We thank the reviewer for this insightful suggestion. To quantify the mutual benefits of joint human-scene modeling, **we conducted an ablation and visualized the cross-attention between human and scene branches** in the revised manuscripts (`Tab.10`, `Fig.20`, and `L.1360-L.1377`).
>
> **1\. Ablation (Added `Tab.10b`)**
> As shown in table below, the model without cross-attention performs worse across all tasks: 3D Reconstruction, Human Mesh Reconstruction (Local Human), and Global Human Motion (Global Human).
>
> * **Benefit from Scene Context (Local Human):** The degradation in Local Human Reconstruction (PA-MPJPE 84.9 vs. 93.9) aligns with our findings in `Appendix A.6`, confirming that Human3R benefits significantly from the 3D geometric cues provided by the scene context.
> * **Metric Alignment (Global Human):** The significant drop is observed in Global Human Motion (WA-MPJPE 267.9 vs. 659.8). This strictly verifies that cross-attention is essential for aligning human and scene representations within a unified metric space.
>
> | Ablation | Scene |  | Local Human |  |  | Global Human |  |  |
> | :---- | :---: | :---: | :---: | :---: | :---: | :---: | :---: | :---: |
> |  | Abs Rel $\\downarrow$ | $\\delta \< 1.25 \\uparrow$ | PA-MPJPE $\\downarrow$ | MPJPE $\\downarrow$ | PVE $\\downarrow$ | WA-MPJPE $\\downarrow$ | W-MPJPE $\\downarrow$ | RTE $\\downarrow$ |
> | Full model | **0.086** | **95.4** | **84.9** | **71.2** | **44.1** | **267.9** | **112.2** | **2.2** |
> | w/o cross attention | 0.089 | 94.5 | 93.9 | 78.9 | 47.1 | 659.8 | 256.3 | 3.7 |
>
> **2\. Visualization of Cross-attention (Added `Fig.19`)**
> We provide the visualization of decoder’s cross-attention between human prompts and image features. These visualizations intuitively illustrate that the network effectively captures correlated human-scene features during reconstruction.
>
> ---
>
> > **Q3: The ViT encoder from CUT3R may downscale spatial resolution, which might influence human detection and segmentation accuracy. Have the authors analyzed how input or feature resolution affects performance? It would be valuable to evaluate this by resizing the encoded features with various (i, j) scaling factors.**
>
> We agree with the reviewer that ViT encoders inherently downscale spatial resolution, which may potentially impact human detection granularity. Following your suggestion, **we evaluate detection performance by resizing encoder features with scaling factors** (ranging from $\\times 1/8$ to $\\times 8$) the revised manuscript (`Tab.11` and `L.1390–L.1403`).
>
> As shown in table below, we report three metrics:
> **1\) Precision**: percentage of detected objects that are valid human;
> **2\) Recall**: percentage of ground-truth humans that are detected;
> **3\) F1-score**: harmonic mean of Precision and Recall.
>
> | Scaling factors | Precision $\\uparrow$ | Recall $\\uparrow$ | F1-score $\\uparrow$ |
> | :---: | :---: | :---: | :---: |
> | $\\times 1/8$ | **99.4** | 55.8 | 63.8 |
> | $\\times 1/4$ | 98.1 | 86.4 | 91.3 |
> | $\\times 1/2$ | 98.8 | 93.7 | 96 |
> | $\\times 1$ | 99.1 | **94.9** | **96.8** |
> | $\\times 2$ | 99 | 94.8 | 96.7 |
> | $\\times 4$ | 99 | 94.8 | 96.7 |
> | $\\times 8$ | 99 | 94.8 | 96.7 |
>
> **Analysis:**
>
> * **Stable precision:** Precision remains consistently high (\>98%) across all scales, indicating that resolution changes do not induce false positives.
> * **Recall degradation at low resolution:** While Recall remains stable at higher resolutions, it drops significantly at lower resolutions (e.g., $\\times 1/8$). This drop occurs because low-res feature maps can cause token collision, where different human heads in close proximity collapse into a single head token at low resolutions. This leads to missing individual instances, thus increasing false negatives.
>
> Consequently, we recommend utilizing higher feature resolutions to ensure robust detection, particularly in crowded scenarios.

---

> ### Author Response · Authors · 2025-11-30
> **Official Response by Authors (3/3)**
>
> > **Q4: In scenarios where multiple humans are heavily occluded and occupy the same head token, how does the model differentiate and handle such cases? A clarification or visualization would be helpful to understand its robustness under severe occlusion.**
>
> Thank you for your careful reviewing and bringing up this valuable point. **We add visualization on multi-human scenes** (`Fig.10`, `L.469–L.485`)**, and a robustness analysis** (`Fig.14`, `L.1003-L.1064`) in the revised manuscript.
>
> **1\. Visualization (Added `Fig.10`):**
> We evaluate OOD multi-person videos for both human and scene reconstruction:
>
> * **Robust human-scene reconstruction:** Human3R robustly estimates 3D scene structures and camera poses even when the view is heavily occluded by humans. It accurately reconstructs observed humans within the global scene with consistent motion trajectories and stable ID tracking.
>
> **2\. Robustness to occlusion (Added `Fig.14`)**
>
> * **Limitations:** As mentioned in Conclusion, we noted that Human3R sometimes struggles to detect humans when the head is not visible, as the head serves as the primary discriminative keypoint. We observe distinguishing subjects becomes challenging when multiple humans are heavily occluded to the extent that they occupy the same head token.
> * **Robustness:** However, the model demonstrates exciting robustness in less extreme cases. It remains capable of detecting humans when:
>   * **Only a small portion** of the head is visible (see `Fig.13 left` and `Fig.14`);
>   * **Anatomical points adjacent to the head** (e.g., chest, back, or neck) are visible (see `Fig.13` right and `Fig.14`).
>
> **Discussion: Training Strategy.** This general robustness stems from our training strategy (following the data augmentation protocol from Multi-HMR):
>
> * We apply image cropping during training.
> * Crucially, if the head is truncated, we supervise the network to identify the visible point closest to the unobserved head center (e.g., the chest or neck).
>
> We hypothesize that incorporating more aggressive cropping augmentations and training on heavily occluded datasets would further enhance robustness and reduce misses.

---

### Official Review · Reviewer_B3ku · 2025-11-02

**Soundness:** 3
**Presentation:** 3
**Contribution:** 3
**Rating:** 8
**Confidence:** 5

**Summary:**

- Humans3R predicts humans, scene, and camera jointly with a single model in a single step.
- It freezes CUT3R (a state-based recurrent 3D scene reconstruction model) and augments it with visual prompt tuning (learned tokens) to predict human mesh parameters. Mesh parameters are predicted only from tokens classified as human; this classifier is learned end-to-end.
- The main benefit is simplicity: the work unifies previously separate techniques for scene and human reconstruction.
- To handle longer time horizons, Humans3R adopts TTT3R for test-time sequence length adaptation.
- The main evaluations are done on 3DPW, EMDB1/2 and RICH datasets.
- Main baselines: WHAM, JOSH3R, Multi-HMR.
- Results: Humans3R outperforms corresponding one-stage online methods on both local and global human mesh reconstruction.

**Strengths:**

- The paper is well written, well organized, and easy to follow; key ideas are presented clearly with intuitive explanations, and the technical details are clear and consistent.

- Humans3R is novel: the first to unify human and scene reconstruction in a single online model with promising results. I am pleasantly surprised to find out that CUT3R-like models can be repurposed (using human priors from the Multi-HMR image encoder) to perform joint scene and human mesh reconstruction. No depth estimation or SLAM needed. This is a strong technical contribution and a promising direction for the community, the resulting simplification is notable and welcome.

- Empirical results are strong and thorough: quantitative evaluations span multiple datasets (3DPW, EMDB, RICH) and method classes (multi-stage, online, offline). Locally (3DPW/EMDB-1), Human3R beats one-stage baselines (e.g., MPJPE 71.2 on 3DPW vs. BEV 78.5); globally (EMDB-2/RICH), it notably improves over online WHAM (e.g., EMDB-2 WA-MPJPE 112.2 vs. 135.6; RTE 2.2% vs. 6.0%).

- Intrinsic robustness via metric-scale scene context. Leveraging CUT3R’s frozen metric-scale scene prior, the system remains stable without calibrated intrinsics and across aspect ratios. It would be valuable to test generalization beyond the reported datasets and qualitative examples; if confirmed, this would address a common failure mode in HMR pipelines.

**Weaknesses:**

- Dependence on the head-token: The authors already note this, but this is a substantial assumption and, in my view, the primary weakness. Humans3R’s forward pass relies on reliable head-token detection; occlusion or truncation can cause misses. The quantitative evaluations do not reflect this limitation because the datasets lack such scenarios, whereas multi-stage baselines perform reasonably under truncation due to strong train-time cropping augmentations.

- Inference on non-full body + multi-person sequences: The reported results focus primarily on full-body sequences. It would be insightful to understand the performance degradation on challenging quarter/upper-body or face-focused images, where the head is still visible but the sequence is out-of-distribution relative to BEDLAM’s training data. How accurate would the scene reconstruction be in these cases? Also specify crowding limits: what is the maximum number of concurrent identities Humans3R handles in practice (for eg, SLAHMR, CVPR 2023 manages about 10 identities offline)?

- Pixel aligned predictions: The qualitative results (and supplementary videos) are primarily 3D visualizations. They are metrically consistent and impressive. Over longer horizons, predicted camera pose may drift, leading to misalignment. It would be helpful to demonstrate mesh–camera consistency by projecting the mesh into the image and overlaying it, and compare against baselines.

**Questions:**

Overall, I like this work and rating the work as 8.

My questions are primarily on the mentioned weaknesses.
- How would you remove head-token dependence in the architecture? Please expand on inclusion of body point localizers.
- How well does Humans3R generalize to non-fully body + multi-person scenarios?
- Mesh projection visualization.

---

> ### Author Response · Authors · 2025-11-30
> **Official Response by Authors (1/3)**
>
> We appreciate the feedback. We revise the manuscript and address each point below:
>
> > **Q1: Dependence on the head-token: The authors already note this, but this is a substantial assumption and, in my view, the primary weakness. Humans3R’s forward pass relies on reliable head-token detection; occlusion or truncation can cause misses. The quantitative evaluations do not reflect this limitation because the datasets lack such scenarios, whereas multi-stage baselines perform reasonably under truncation due to strong train-time cropping augmentations.**
>
> Thank you for your valuable suggestion. We agree with the reviewer that the current quantitative evaluations do not fully reflect limitations regarding truncation and occlusion; therefore, **we have included a qualitative robustness analysis** in the revised manuscript (`Fig.13`, `Fig.14` and `L.1003–L.1064`). Our observations are summarized below:
>
> **Limitation:** As mentioned in Conclusion, we noted that Human3R sometimes struggles to detect humans when the head is not visible due to severe truncation or occlusion, as the head serves as the primary discriminative keypoint.
>
> **Robustness (Added** `Fig.13`, `Fig.14`**):** However, the model demonstrates exciting robustness in less extreme cases. It remains capable of detecting humans when:
>
> * **Only a small portion** of the head is visible (see `Fig.13 left` and `Fig.14`);
> * **Anatomical points adjacent to the head** (e.g., chest, back, or neck) are visible (see `Fig.13 right` and `Fig.14`).
>
> **Discussion: Training Strategy.** This general robustness stems from our training strategy (following the data augmentation protocol from Multi-HMR):
>
> * We apply image cropping during training.
> * Crucially, if the head is truncated, we supervise the network to identify the visible point closest to the unobserved head center (e.g., the chest or neck).
>
> We hypothesize that incorporating more aggressive cropping augmentations and training on heavily occluded datasets would further enhance robustness and reduce misses.

---

> ### Author Response · Authors · 2025-11-30
> **Official Response by Authors (2/3)**
>
> > **Q2: Inference on non-full body \+ multi-person sequences: The reported results focus primarily on full-body sequences. It would be insightful to understand the performance degradation on challenging quarter/upper-body or face-focused images, where the head is still visible but the sequence is out-of-distribution relative to BEDLAM’s training data. How accurate would the scene reconstruction be in these cases? Also specify crowding limits: what is the maximum number of concurrent identities Humans3R handles in practice (for eg, SLAHMR, CVPR 2023 manages about 10 identities offline)?**
>
> We thank the reviewer for carefully reviewing our paper and bringing up this insightful comment. **We follow your suggestion to test generalization beyond the reported datasets** in the revised manuscript (`Fig.10`, `Fig.15`, `Fig.16`, and `L.432–L.485`, `L.1067–L.1117`).
>
> **1\. Non-full body sequences (Added `Fig.15`, `Fig.16`)**
> Since standard ground-truth benchmarks lack non-full body data, we tested on challenging upper-body views, face close-ups, and OOD poses:
>
> * **Success cases (Plausible inference):** As shown in **`Fig.15`**, Human3R reconstructs consistent 3D scenes and camera poses even when the image is mostly occupied by the upper body. By leveraging a strong learned scene-aware pose prior, our model not only reconstructs the observable upper body but also plausibly infers the unobserved lower body, yielding poses that stand naturally on the reconstructed ground plane.
> * **Limitations (Pose ambiguity & Extreme cases):** We observe limitations in specific OOD scenarios (**`Fig.16`**):
>   * *Pose Ambiguity:* In face-focused running sequences (`Fig.16 left`), the model correctly reconstructs the scene and camera trajectory but defaults to a standing pose. This reveals a limitation of our deterministic pipeline, which tends to regress to the mean pose when visual evidence is missing. Integrating generative modeling to capture multimodal distributions represents a promising direction for future work.
>   * *Fine-grained Details & Extreme Poses:* Performance degrades on fine-grained scene reconstruction (e.g., wall clock, `Fig.16 middle`), unobserved extremities (e.g., hands in selfies, `Fig.16 middle`), or extreme OOD poses (e.g., breakdancing, `Fig.16 right`). Addressing these limitations may require scaling up the training dataset and exploring self-supervised solutions.
>
> **2\. Multi-person sequences (Added `Fig.10`)**
> Current quantitative evaluations do not reflect performance in crowded scenarios, as ground-truth typically contains only one (EMDB, RICH) or two people (3DPW). Thus we qualitatively evaluated OOD multi-person videos for both human and scene reconstruction:
>
> * **Robust reconstruction:** Human3R robustly estimates 3D scene structures and camera poses even when the view is heavily occluded by humans. It accurately reconstructs observed humans within the global scene with consistent motion trajectories and stable ID tracking.
> * **Efficiency on crowds:** A key advantage of Human3R is its one-shot inference. Unlike top-down methods (e.g., SLAHMR) or per-person HMRs that scale linearly with the number of people, our bottom-up approach maintains constant inference speed regardless of crowd density. Crucially, Human3R extends this capability to reconstruct global humans jointly with the 3D scene in a single feed-forward pass.
> * **Limitations:** The practical limit is not a fixed number of identities, but rather the spatial resolution of head tokens. The model struggles to differentiate subjects when multiple humans are heavily occluded to the extent that they occupy the same head token (discussed in `Appendix A.1`).

---

> ### Author Response · Authors · 2025-11-30
> **Official Response by Authors (3/3)**
>
> > **Q3: Pixel aligned predictions: The qualitative results (and supplementary videos) are primarily 3D visualizations. They are metrically consistent and impressive. Over longer horizons, predicted camera pose may drift, leading to misalignment. It would be helpful to demonstrate mesh–camera consistency by projecting the mesh into the image and overlaying it, and compare against baselines.**
>
> Thanks for your observation\! **We have further evaluated mesh projection and compared against baselines** in the revised manuscript (`Fig.17`, `L.1119-L.1131`).
>
> **Comparison with Baselines (Add `Fig.17`).** As illustrated in `Fig.17`, we compare our method against GVHMR and TRAM on long sequences from 3DPW dataset:
>
> * **GVHMR** produces superior mesh projection aligned the body contour well but is limited to single-person estimation (typically the largest pixel-occupied one), failing in complex multi-person occlusion.
> * **TRAM** exhibits slightly less accurate pixel-aligned mesh projection than GVHMR, but natively models multi-person HMR leading to robust results in crowded scenes.
>
> The results demonstrate that our method **successfully maintains mesh-camera consistency over long horizons without significant drift**. We achieve pixel-aligned accuracy comparable to GVHMR while retaining the robust multi-person capability as TRAM (also shown in `Sec.4.4`).
>
> ---
>
> > **Q4: How would you remove head-token dependence in the architecture? Please expand on inclusion of body point localizers.**
>
> **We provide a discussion on how to expand on body point localizers** in the revised manuscript (`L.1058-L.1064`). Specifically, we discuss two possible pathways to eliminate the single-point dependency:
>
> **1\.  Pixel-aligned body point localizers:**
> Instead of a single head token, future work can employ pixel-aligned body point localizers (NLF, DualPM) to localize multiple canonical body points in pixel space to serve as a diverse set of prompts. This could be followed by regressing the corresponding SMPL parameters and a post-processing step to merge multiple queries.
>
> **2\.  Region-based prompts:**
> Following the success of PromptHMR, another promising avenue is to utilize bounding boxes or segmentation maps as prompts. Unlike specific keypoints, region-based prompts operate on any observable body region. This allows the model to identify and reconstruct humans even when canonical keypoints are occluded, significantly enhancing robustness in crowded or truncated scenes.

---

### Author Response · Authors · 2025-11-30
**General Response (3/3)**

## Part III. To R3 (N5HW): Summary of Clarifications

We specifically thank R3 for the detailed technical review. We noticed some concerns stemmed from misunderstandings regarding the **inference pipeline scope** and **architectural differences**. We summarize our targeted responses below:

| Key Concerns from R3 | Our Revision | Key Rebuttal Points |
| :---- | :---- | :---- |
| **1\. Similarity to GVHMR** *(Similar concepts & complexity?)* | Added runtime breakdown: `Tab.5`,  `Tab.6`,  `Tab.7` | **Offline vs. Online.** GVHMR relies on external modules (SLAM, YOLO) for **Single-person**;  Human3R is an **end-to-end** & **online** framework for **Multi-person \+ 3D Scene**. |
| **2\. Real-time Claim** *(Reported slower speed in test?)* | Added speed benchmark: `Tab.4`,  `Tab.8` | **Validated 15.8 FPS.** Perceived latency stemmed from *evaluation overhead (GT process, metrics, visualization, I/O)*, not model inference. We will release `inference_only.py` for reproducibility of the raw model speed. Our lightweight model runs **real-time** on RTX 4090 across 6 datasets. |
| **3\. Joint-Train Decoder** *(Why not fine-tune decoder?)* | Added decoder ablation: `Tab.10c` | **Justified Parameter Efficiency.** Joint training causes **catastrophic forgetting** of scene priors (Error explodes: 267 $\\to$ 1359), verifying our **human prompt tuning** design. |
| **4\. Source of Gain** *(Improvement in camera coordinate comes from depth or other dims?)*  | Added improvement analysis: `Tab.9`,  `Fig.18` | **Benefits from Scene Context.**  Drastically reduces depth error (**\~60%** MRPE), refines **pose/shape fidelity** (**\~20%** PVE), and improves **OOD robustness** (**\~50%** PVE while intrinsic-free).  |
| **5\. Token Clarity** *(Ambiguity on feature extraction)* | New visualization: `Fig.6`; Revised caption: `L.223` | **Clarified Dual-Encoder.** Explicitly visualized the integration of separate **Multi-HMR** (human) and **CUT3R** (scene) encoders. |

For more details, we provide point-by-point responses in each reviewer’s thread.

---

### Author Response · Authors · 2025-11-30
**General Response (2/3)**

## Part II. Key Highlights from New Experiments

To address concerns regarding differences, efficiency, ablations, and robustness, we conduct comprehensive new experiments. We summarize the three most key findings below:

### **1\. Differences & Efficiency (R2, R3)**

*Responding to conceptual differences (R3) and efficiency comparisons (R2) vs. baselines:*
We clarify that Human3R can directly be used for **online inference** and **multi-person \+ 3D scenes**.

* **Conceptual Differences:** GVHMR and TRAM can only estimate single human and multiple humans respectively, and no scene.

* **Efficiency:** Human3R does not require external modules (e.g., Detectors, Tracking, 2D keypoint estimator, SLAM), making it an **end-to-end and scalable** method.

| Methods | GVHMR | TRAM | Human3R |
| :---- | :---: | :---: | :---: |
| 3D scene | ✗ | ✗ | **✓** |
| Multi-person | ✗ | **✓** | **✓** |
| End-to-end learning | ✗ | ✗ | **✓** |
| Online | ✗ | ✗ | **✓** |
| FPS (RTX 4090\) | \~5 | \~1 | **5\~15** |

To further justify our real-time claim, we profile different backbones. Human3R offers configurable scalability:

| Backbone | FPS (RTX 4090\) | Global Human (WA-MPJPE$\\downarrow$) | Use Case |
| :---- | :---: | :---: | :---- |
| **ViT-S/672** | **15.87** | 129.9 | **Real-time Speed** |
| ViT-B/672 | 13.33 | 122.1 | Balanced |
| **ViT-L/896** | 5.38 | **112.2** | **More Accurate Reconstruction** |

---

&nbsp;

### **2\. Ablation & Source of Gain (R2, R3)**

*Reviewers suggested analysis on cross-attention (R2) and improvements in camera coordinates (R3).*

**A. Necessity of Cross-Attention:** Removing it degrades performance, confirming joint modeling aligns humans in the global scene.

| Ablation | Local Human (PA-MPJPE$\\downarrow$) | Global Human (WA-MPJPE$\\downarrow$)  | Analysis |
| :---- | :---: | :---: | :---- |
| w/o Cross-Attn | 93.9 | 659.8 | Fails to align human & scene |
| **Full Model** | **84.9** | **267.9** | **Corrects global placement** |

**B. Why Scene Matters:** Incorporating scene context yields holistic improvements: it drastically reduces absolute **depth & location error** (MRPE), refines **human pose & shape fidelity** (PVE), and ensures **robustness** without requiring camera intrinsics, surpassing Multi-HMR even when it uses Ground-Truth intrinsics.

| Method | Pose: PVE$\\downarrow$ | Depth: MRPE$\\downarrow$ | Note |
| :---- | :---: | :---: | :---- |
| Multi-HMR (w/o GT intrinsic) | 77.0 | 1347.5 | High depth ambiguity |
| Multi-HMR (w/ **GT intrinsic**) | 64.1 | 559.9 | Requires known camera |
| **Human3R (Ours)** | **62.2** | **212.1** | **Scene-aware & Intrinsic-free** |

---

&nbsp;

### **3\. Robustness Analysis (R1, R2)**

We add new qualitative results on challenging OOD scenarios:

| Challenge | Visualization | Conclusion |
| :---- | :---: | :---- |
| **Head Truncation / Occlusion (R1, R2)** | `Fig.13`, `Fig.14` | **Robust Detection:**  Localizes via adjacent parts (neck/chest) when partial occlusion. |
| **Non-Full Body (R1)** | `Fig.15`, `Fig.16` | **Plausible Inference:**  Recovers full-body poses & scenes despite severe occlusion. |
| **Crowded Scenes (R1, R2)** | `Fig.10` | **Generalizable:**  Handles **\>10 people** via efficient **one-shot** inference. |

---

### Author Response · Authors · 2025-11-30
**General Response (1/3)**

We thank **R1(B3ku)**, **R2(HBKC)**, **R3(N5HW)** for their constructive feedback. We are encouraged by their consensus on the contribution of our work. Human3R code will be released for future research.

Common recognition of our strengths includes:

1. *Novelty & Contribution:*
   Recognized as a **"strong technical contribution"** (R1) and the **"first to unify"** online human-scene reconstruction (R1). R1 was **"pleasantly surprised"** by the approach, which **"advances integration"** (R3) even under **"limited training data"** (R3).

2. *Strong Performance:*
   Experiments reveal **"strong and thorough"** (R1), **"consistently improved"** (R2) results and **"valuable insights"** (R3) via **"insightful"** ablations (R2) and **"promising and engaging"** visualizations (R3).

3. *Simplicity & Potential:*
   A **"simple yet powerful"** idea (R2), offering a **"notable simplification"** (R1) that removes heavy dependencies, represents a **"promising direction for the community"** (R1).

We have conducted extensive additional experiments (added **9 new analysis subsections in Appendix**) to address each concern, and revised the manuscript accordingly.

All changes in the updated `PDF` are highlighted in **red**.

---

&nbsp;

## Part I: Summary of Revisions

1. **Efficiency Benchmarking (R2, R3)**
    * **Runtime/latency comparison** against baselines (GVHMR, TRAM) (R2-Q1, R3-Q1; `Tab.5`, `Tab.6`, `Tab.7`).
    * **Efficiency evaluation** across all backbones and 6 datasets with varying resolutions (R3-Q3/Q5; `Tab.4`, `Tab.8`).

2. **Component Analysis (R2, R3)**
    * **Ablations** on cross-attention (R2-Q2; `Tab.10b`, `Fig.20`), and joint-train decoder (R3-Q1; `Tab.10c`).
    * **In-depth analysis** of improvements in camera coordinates (R3-Q4; `Tab.9`, `Fig.18`), and encoder feature resolution (R2-Q3; `Tab.11`),

3. **Robustness Analysis (R1, R2)**
    * **Generalization** on severe truncation/occlusion (R1-Q1, R2-Q4; `Fig.13`, `Fig.14`), non-full body captures (R1-Q2; `Fig.15`, `Fig.16`), and crowded scenes (R1-Q2, R2-Q4; `Fig.10`).
    * **Mesh consistency** comparisons against baselines (R1-Q3; `Fig.17`).

4. **Discussions (R1, R3)**
    * Expand discussion on inclusion of body point localizers (R1-Q4; `L.1058`).
    * Clarification on conceptual differences with GVHMR (R3-Q1; `L.1073`), token extraction details (R3-Q2; `Fig.6`, `L.223`), and ambiguous statements (R3-Q5; `L.071`)

---

### Meta-Review · Area_Chair_VBiL · 2026-01-14

**Summary:**

The decision to accept Human3R is informed by the reviewers' constructive critique regarding the model's robustness, efficiency claims, and architectural clarity, all of which the authors addressed substantially in their rebuttal. A primary concern raised by Reviewer B3ku was the framework's reliance on head tokens, questioning its ability to handle severe truncation or occlusion where the head is not visible. Similarly, both Reviewers B3ku and HBKC requested more rigorous evaluation on out-of-distribution scenarios, such as non-full body sequences and crowded scenes, to verify the model's generalization capabilities beyond standard benchmarks. On the technical side, Reviewers HBKC and N5HW scrutinized the paper's real-time performance claims. Additionally, Reviewer N5HW questioned the method's novelty relative to GVHMR. The authors' extensive rebuttal, including new robustness analysis, detailed runtime breakdowns, and ablations validating the "human prompt tuning" design, effectively resolved these concerns, solidifying the consensus on the paper's contribution. Therefore, acceptance is recommended.

**Reviewer Concerns:**

The rebuttal successfully addressed concerns regarding the model’s efficiency claims and architectural validity. The authors provided extensive runtime benchmarks to substantiate the real-time (15 FPS) capability, clarifying that previous discrepancies arose from evaluation overhead rather than inference latency. They also effectively distinguished their work from GVHMR by demonstrating their method's unique online, multi-person, and scene-aware capabilities. Furthermore, the ablation study on the joint-train decoder justified their design choice, empirically proving that full fine-tuning causes catastrophic forgetting of scene priors.

However, concerns regarding the model's fundamental reliance on head tokens remain partially outstanding. While the authors demonstrated robustness to partial occlusion, they explicitly acknowledged that the system still fails when the head is fully invisible or in crowded scenes where "head token collision" occurs. The proposed solutions, such as using body-point localizers or region-based prompts, were discussed only as directions for future work. Additionally, the limitation of deterministic inference persists, as the model reverts to a mean pose during ambiguous, face-focused sequences.

**Reviewer Scores:**

Reviewer B3ku initially gave an 8, praising the unified approach. Since the authors effectively addressed concerns about head-token dependence and mesh drift with new visualizations and training clarifications, B3ku would likely have maintained their strong acceptance rating, potentially increasing confidence in the model's robustness.

Reviewer HBKC also started with an 8. The authors provided the requested runtime comparisons against GVHMR/TRAM and cross-attention ablations. Satisfied by the evidence of efficiency and component necessity, HBKC would almost certainly have kept their high score, seeing the "real-time" claims fully vindicated.

Reviewer N5HW gave a 4, citing concerns about similarity to GVHMR and doubted real-time performance. The rebuttal clarified fundamental differences (online vs. offline) and proved 15 FPS capabilities. With these misunderstandings resolved, N5HW would likely have raised their score to a 6.

---

### Decision · Program_Chairs · 2026-01-26

Accept (Poster)